# The Exact Sample Complexity Gain from Invariances for Kernel Regression

**Behrooz Tahmasebi**
MIT CSAIL
bzt@mit.edu

**Stefanie Jegelka**
MIT CSAIL and TU Munich
stefje@mit.edu

## Abstract

In practice, encoding invariances into models improves sample complexity. In this work, we study this phenomenon from a theoretical perspective. In particular, we provide minimax optimal rates for kernel ridge regression on compact manifolds, with a target function that is invariant to a group action on the manifold. Our results hold for any smooth compact Lie group action, even groups of positive dimension. For a finite group, the gain effectively multiplies the number of samples by the group size. For groups of positive dimension, the gain is observed by a reduction in the manifold's dimension, in addition to a factor proportional to the volume of the quotient space. Our proof takes the viewpoint of differential geometry, in contrast to the more common strategy of using invariant polynomials. This new geometric viewpoint on learning with invariances may be of independent interest.

## 1 Introduction

In a broad range of applications, including machine learning for physics, molecular biology, point clouds, and social networks, the underlying learning problems are invariant with respect to a group action. The invariances are observed widely in practice, for instance, in the study of high energy particle physics [21, 32], galaxies [23, 15, 2], and also molecular datasets [1, 59, 36] (see [67] for a survey). In learning with invariances, one aims to develop powerful architectures that exploit the problem's invariance structure as much as possible. An essential question is thus: what are the fundamental benefits of model invariance, e.g., in terms of sample complexity?

Several architectures for learning with invariances have been proposed for various types of data and invariances, including DeepSet [70] for sets, Convolutional Neural Networks (CNNs) [28], PointNet [53, 54] for point clouds with permutation invariance, tensor field neural networks [62] for point clouds with rotations, translations, and permutations symmetries, Graph Neural Networks (GNNs) [58], and SignNet and BasisNet [37] for spectral data. Other works study invariance with respect to the orthogonal group [63], and invariant and equivariant GNNs [42]. These architectures are to exploit the invariance of data as much as possible, and are invariant/equivariant by design.

In fixed dimensions, one common feature of many invariant models, including those discussed above, is that the data lie on a compact manifold (not necessarily a sphere, e.g., the Stiefel manifold for spectral data), and are invariant with respect to a group action on that manifold. Thus, characterizing the theoretical gain of invariances corresponds to studying the gain of learning under group actions on manifolds. Adopting this view, in this paper, we answer the question: *how much gain in sample complexity is achievable by encoding invariances?* As this problem is algorithm and model dependent, it is hard to address in general. A focused version of the problem, but still interesting, is to study this sample complexity gain in kernel-based algorithms, which is what we address here. As neural networks in certain regimes behave like kernels (for example, the Neural Tangent Kernel (NTK) [26, 34]), the results on kernels should be understood as relevant to a range of models.

37th Conference on Neural Information Processing Systems (NeurIPS 2023).

Formally, we consider the Kernel Ridge Regression (KRR) problem with i.i.d. data on a compact manifold $\mathcal{M}$. The target function lies in a Reproducing Kernel Hilbert space (RKHS) of Sobolev functions $\mathcal{H}^s(\mathcal{M})$, $s \geq 0$. In addition, the target function is invariant to the action of an arbitrary Lie group $G$ on the manifold. We aim to quantify: *by varying the group $G$, how does the sample complexity change, and what is the precise gain as $G$ grows?*

**Main results.** Our main results characterize minimax optimal rates for the convergence of the (excess) population risk (generalization error) of KRR with invariances. More precisely, for the Sobolev kernel, the most commonly studied case of kernel regression, we prove that a (excess) population risk (generalization error) $\propto \left( \frac{\sigma^2 \, \mathrm{vol}(\mathcal{M}/G)}{n} \right)^{s/(s+d/2)}$ is both achievable and minimax optimal, where $\sigma^2$ is the variance of the observation noise, $\mathrm{vol}(\mathcal{M}/G)$ is the volume[1] of the corresponding quotient space, and $d$ is the effective dimension of the *quotient space* (see Section 4 for a precise definition). This result shows a reduction in sample complexity in *two* intuitive ways: (1) scaling the effective number of samples, and (2) reducing dimension and hence exponent. First, for finite groups, the factor $\mathrm{vol}(\mathcal{M}/G)$ reduces to $\mathrm{vol}(\mathcal{M})/|G|$, and can hence be interpreted as scaling the *effective* number of samples by the size of the group. That is, each data point conveys the information of $|G|$ data points due to the invariance. Second, and importantly, the parameter $d$ in the exponent can generally be much smaller than $\dim(\mathcal{M})$, which would be the correspondent of $d$ in the non-invariant case. In the best case, $d = \dim(\mathcal{M}) - \dim(G)$, where $\dim(G)$ is the dimension of the Lie group $G$. Hence, the second gain shows a gain in the dimensionality of the space, and hence in the exponent.

Our results generalize and greatly expand previous results by Bietti et al. [5], which only apply to *finite* groups and *isometric* actions and are valid only on spheres. In contrast, we derive optimal rates for all compact manifolds and smooth compact Lie group actions (not necessarily isometric), including groups of positive dimension. In particular, the reduction in dimension applies to infinite groups, since for finite groups $\dim(G) = 0$. Hence, our results reveal a new perspective on the reduction in sample complexity that was not possible with previous assumptions. Our rates are consistent with the classical results for learning in Sobolev spaces on manifolds without invariances [24]. To illustrate our general results, in Section 5, we make them explicit for kernel counterparts of popular invariant models, such as DeepSets, GNNs, PointNet, and SignNet.

Even though our theoretical results look intuitively reasonable, the proof is challenging. We study the space of invariant functions as a function space on the quotient space $\mathcal{M}/G$. To bound its complexity, we develop a dimension counting theorem for functions on the quotient space, which is at the heart of our analysis and of independent interest. The difficulty is that $\mathcal{M}/G$ is not always a manifold. Moreover, it may exhibit non-trivial boundaries that require boundary conditions to study function spaces. Different boundary conditions can lead to very different function spaces, and a priori the appropriate choice for the invariant functions is unclear. We prove that smooth invariant functions on $\mathcal{M}$ satisfy the Neumann boundary condition on the (potential) boundaries of the quotient space, thus characterizing exactly the space of invariant functions.

The ideas behind the proof are of potential independent interest: we provide a differential geometric viewpoint of the class of functions defined on manifolds and study group actions on manifolds from this perspective. This stands in contrast to the classical strategy of using polynomials generating the class of functions [46, 5], which is restricted to spheres. To the best of our knowledge, the tools used in this paper are new to the literature on learning with invariances.

In short, in this paper we make the following contributions:

- We characterize the exact sample complexity gain from invariances for kernel regression on compact manifolds for an arbitrary Lie group action. Our results reveal two ways to reduce sample complexity, including a new reduction in dimensionality that was not obtainable with assumptions in prior work.

- Our proof analyzes invariant functions as a function space on the quotient space; this differential geometric perspective and our new dimension counting theorem, which is at the heart of our analysis, may be of independent interest.

---

[1]The quotient space is not a manifold, but one can still define a notion of volume for it; see Section 4.

## 2 Related Work

The perhaps closest related work to this paper is [5], which considers the same setup for finite isometric actions on spheres. We generalize their result in several aspects: the group actions are not necessarily finite or isometric, and the compact manifold is arbitrary (including compact submanifolds of $\mathbb{R}^d$), allowing to observe a new axis of complexity gain. Mei et al. [46] consider invariances for random features and kernels, but in a different scaling/regime; thus, theirs are not comparable to our results. For density estimation on manifolds, optimal rates are given in [24], which are consistent with our theory. McRae et al. [45] show non-asymptotic sample complexity bounds for regression on manifolds. A similar technique was recently applied in [40], but for a very different setting of covariate shifts.

The generalization benefits for invariant classifiers are observed in the most basic setup in [61], and for linear invariant/equivariant networks in [20, 19]. Some works propose covering ideas to measure the generalization benefit of invariant models [71], while others use properties of the quotient space [57, 50]. It is known that structured data exhibit certain gains for localized classifiers [14]. Sample complexity gains are also observed for CNN on images [17, 35]. Wang et al. [66] incorporate more symmetry in CNNs to improve generalization.

Many works introduce models for learning with invariances for various data types; in addition to those mentioned in the introduction, there are, e.g., group invariant scattering models [41, 9]. A probabilistic viewpoint of invariant functions [6] and a functional perspective [72] also exist in the literature. The connection between group invariances and data augmentation is addressed in [13, 39].

Universal expressiveness has been studied for settings like rotation equivariant point clouds [18], sets with symmetric elements [44], permutation invariant/equivariant functions [56], invariant neural networks [55, 69, 43], and graph neural networks [68, 49]. Lawrence et al. [30] study the implicit bias of linear equivariant networks. For surveys on invariant/equivariant neural networks, see [22, 8].

## 3 Preliminaries and Problem Statement

Consider a smooth connected compact boundaryless[2] $\dim(\mathcal{M})$-dimensional (Riemannian) manifold $(\mathcal{M}, g)$, where $g$ is the Riemannian metric. Let $G$ denote an arbitrary compact Lie group of dimension $\dim(G)$ (i.e., a group with a smooth manifold structure), and assume that $G$ acts smoothly on the manifold $(\mathcal{M}, g)$; this means that each $\tau \in G$ corresponds to a diffeomorphism $\tau : \mathcal{M} \to \mathcal{M}$, i.e., a smooth bijection. Without loss of generality, we can assume that $G$ acts *isometrically* on $(\mathcal{M}, g)$, i.e., $G$ is a Lie subgroup of the isometry group $\mathrm{ISO}(\mathcal{M}, g)$. To see why this is not restrictive, given a base metric $g$, consider a new metric $\tilde{g} = \mu_G(\tau^* g)$, where $\mu_G$ is the left-invariant Haar (uniform) measure of $G$, and $\tau^* g$ is the pullback of the metric $g$ by $\tau$. Under the new metric, $G$ acts isometrically on $(\mathcal{M}, \tilde{g})$. We review basic facts about manifolds and their isometry groups in Appendix A.1 and Appendix A.2.

We are given a dataset $\mathcal{S} = \{(x_i, y_i) : i = 1, 2, \ldots, n\} \subseteq (\mathcal{M} \times \mathbb{R})^n$ of $n$ labeled samples, where $x_i \sim_{\text{i.i.d.}} \mu$, for the uniform (Borel) probability measure $d\mu(x) := \frac{1}{\mathrm{vol}(\mathcal{M})} d\mathrm{vol}_g(x)$. Here, $d\mathrm{vol}_g(x)$ denotes the volume element of the manifold defined using the Riemannian metric $g$. We assume the uniform sampling for simplicity; our results hold for non-uniform cases, too. The hypothesis class is a set $\mathcal{F} \subseteq L^2_{\text{inv}}(\mathcal{M}, G) \subseteq L^2(\mathcal{M})$ including only $G$-invariant square-integrable functions on the manifold, i.e., those $f \in L^2(\mathcal{M})$ satisfying $f(\tau(x)) = f(x)$ for all $\tau \in G$. We assume that there exists a function $f^\star \in \mathcal{F}$ such that $y_i = f^\star(x_i) + \epsilon_i$ for each $(x_i, y_i) \in \mathcal{S}$, where $\epsilon_i$'s are conditionally zero-mean random variables with variance $\sigma^2$, i.e., $\mathbb{E}[\epsilon_i | x_i] = 0$ and $\mathbb{E}[\epsilon_i^2 | x_i] \leq \sigma^2$.

Let $K : \mathcal{M} \times \mathcal{M}$ denote a continuous positive-definite symmetric (PDS) kernel on the manifold $\mathcal{M}$, and let $\mathcal{H} \subseteq L^2(\mathcal{M})$ denote its Reproducing Kernel Hilbert Space (RKHS). The kernel $K$ is called

---

[2]Although the results in this paper can be easily extended to manifolds with boundaries, for simplicity, we focus on the boundaryless case.

*G-invariant* [3] if and only if for all $x, y \in \mathcal{M}$,

$$K(x, y) = K(\tau(x), \tau'(y)), \tag{1}$$

for any $\tau, \tau' \in G$. In other words, $K(x, y) = K([x], [y])$, where $[x] := \{\tau(x) : \tau \in G\}$ is the orbit of the group action that includes $x$.

The Kernel Ridge Regression (KRR) problem on the data $\mathcal{S}$ with a $G$-invariant kernel $K$ asks for the function $\hat{f}$ that minimizes

$$\hat{f} := \underset{f \in \mathcal{H}}{\arg\min} \left\{ \hat{\mathcal{R}}(f) := \frac{1}{n} \sum_{i=1}^{n} (y_i - f(x_i))^2 + \eta \|f\|_{\mathcal{H}}^2 \right\}. \tag{2}$$

By the representer theorem [48], the optimal solution $\hat{f} \in \mathcal{H}$ is of the form $\hat{f} = \sum_{i=1}^{n} a_i K(x_i, .)$ for a weight vector $\mathbf{a} \in \mathbb{R}^n$. The objective function $\hat{\mathcal{R}}(\hat{f})$ can thus be written as

$$\hat{\mathcal{R}}(\hat{f}) = \frac{1}{n} \|\mathbf{y} - \mathbf{K}\mathbf{a}\|_2^2 + \eta \mathbf{a}^T \mathbf{K} \mathbf{a}, \tag{3}$$

where $\mathbf{y} = (y_1, y_2, \ldots, y_n)^n$ and $\mathbf{K} = \{K(x_i, x_j)\}_{i,j=1}^{n}$ is the Gram matrix. This gives the closed form solution $\mathbf{a} = (\mathbf{K} + n\eta I)^{-1} \mathbf{y}$. Using the population risk $\mathcal{R}(f) := \mathbb{E}_{x \sim \mu}[(y - f(x))^2]$, the *effective ridge regression estimator* is defined as

$$\hat{f}_{\text{eff}} := \underset{f \in \mathcal{H}}{\arg\min} \left\{ \mathcal{R}(f) + \eta \|f\|_{\mathcal{H}}^2 \right\}. \tag{4}$$

This paper focuses on the RKHS of Sobolev functions, $\mathcal{H} = \mathcal{H}_{\text{inv}}^s(\mathcal{M}) = \mathcal{H}^s(\mathcal{M}) \cap L_{\text{inv}}^2(\mathcal{M}, G)$, $s \geq 0$. This includes all functions having square-integrable derivatives up to order $s$. Note that $\mathcal{H}^s(\mathcal{M})$ includes only continuous functions when $s > \dim(\mathcal{M})/2$. Moreover, it contains only continuously differentiable functions up to order $k$ when $s > \dim(\mathcal{M})/2 + k$ (Appendix A.10). Note that $\mathcal{H}^s(\mathcal{M})$ is an RKHS if and only if $s > \dim(\mathcal{M})/2$.

## 4 Main Results

Our first theorem provides an upper bound on the excess population risk, or the generalization error, of KRR with invariances.

**Theorem 4.1** (Convergence rate of KRR with invariances). *Consider KRR with invariances with respect to a Lie group $G$ on the Sobolev space $\mathcal{H}_{\text{inv}}^s(\mathcal{M})$, $s > d/2$, with $d = \dim(\mathcal{M}/G)$. Assume that $f^\star \in \mathcal{H}_{\text{inv}}^{s\theta}(\mathcal{M})$ for some $\theta \in (0, 1]$, and let $s = \frac{d}{2}(\kappa + 1)$ for a positive $\kappa$. Then,*

$$\mathbb{E}\left[ \mathcal{R}(\hat{f}) - \mathcal{R}(f^\star) \right] \leq 32 \left( \frac{1}{\kappa\theta} \frac{\omega_d}{(2\pi)^d} \frac{\sigma^2 \operatorname{vol}(\mathcal{M}/G)}{n} \right)^{\theta s/(\theta s + d/2)} \|f^\star\|_{\mathcal{H}_{\text{inv}}^{s\theta}(\mathcal{M})}^{d/(\theta s + d/2)}, \tag{5}$$

*with the optimal regularization parameter*

$$\eta = \left( \frac{1}{2\kappa\theta \|f^\star\|_{\mathcal{H}_{\text{inv}}^{s\theta}(\mathcal{M})}^2} \frac{\omega_d}{(2\pi)^d} \frac{\sigma^2 \operatorname{vol}(\mathcal{M}/G)}{n} \right)^{\theta s/(\theta s + d/2)}, \tag{6}$$

*where $\omega_d$ is the volume of the unit ball in $\mathbb{R}^d$.*

Theorem 4.1 allows to estimate the gain in sample complexity from making the hypothesis class invariant. Setting $G = \{\text{id}_G\}$ (i.e., the trivial group) recovers the standard generalization bound without group invariances. In particular, without invariances, the dimension $d$ becomes $\dim(\mathcal{M})$, and the volume $\operatorname{vol}(\mathcal{M}/G)$ becomes $\operatorname{vol}(\mathcal{M})$. Hence, group invariance can lead to a two-fold gain:

- **Exponent**: the dimension $d$ in the exponent can be much smaller than the corresponding $\dim(\mathcal{M})$.

---

[3] This definition is stronger from being shift-invariant; a kernel $K$ is shift-invariant (with respect to the group $G$) if and only if $K(x, y) = K(\tau(x), \tau(y))$, for all $x, y \in \mathcal{M}$, and all $\tau$. For any given shift-invariant kernel $K$, one can construct its corresponding $G$-invariant kernel $\tilde{K}(x, y) = \int_G K(\tau(x), y) d\mu_G(\tau)$, where $\mu_G$ is the left-invariant Haar (uniform) measure of $G$.

- **Effective number of samples**: the number of samples is multiplied by

$$\frac{\omega_{\dim(\mathcal{M})}/(2\pi)^{\dim(\mathcal{M})}}{\omega_d/(2\pi)^d} \cdot \frac{\mathrm{vol}(\mathcal{M})}{\mathrm{vol}(\mathcal{M}/G)}. \tag{7}$$

The quantity (7) reduces to $|G|$ if $G$ is a finite group that efficiently acts on $\mathcal{M}$ (i.e., if any group element acts non-trivially on the manifold). Intuitively, any sample conveys the information of $|G|$ data points, which can be interpreted as having effectively $n \times |G|$ samples (compared to non-invariant KRR with $n$ samples). For groups of positive dimension, it measures how the group is contracting the volume of the manifold. Note that for finite groups, one always has $\frac{\mathrm{vol}(\mathcal{M})}{\mathrm{vol}(\mathcal{M}/G)} \geq 1$.

**Dimension and volume for quotient spaces.** In Theorem 4.1, the quotient space $\mathcal{M}/G$ is defined as the set of all orbits $[x] := \{\tau(x) : \tau \in G\}$, $x \in \mathcal{M}$, but $\mathcal{M}/G$ is not always a (boundaryless) manifold (Appendix A.5 and A.6). Thus, it is not immediately possible to define its dimension and volume. The quotient space is a finite disjoint union of manifolds, each with its specific dimension/volume. In Appendix A.5 and A.6, we review the theory of quotients of manifolds, and observe that there exists an open dense subset $\mathcal{M}_0 \subseteq \mathcal{M}$ such that $\mathcal{M}_0/G$ is open and dense in $\mathcal{M}/G$, and more importantly, it is a connected precompact manifold. $\mathcal{M}_0/G$ is called the *principal* part of the quotient space. It has the largest dimension among all the manifolds that make up the quotient space.

The projection map $\pi : \mathcal{M}_0 \to \mathcal{M}_0/G$ induces a metric on $\mathcal{M}_0/G$ and this allows us to define $\mathrm{vol}(\mathcal{M}/G) := \mathrm{vol}(\mathcal{M}_0/G)$. Note that $\mathrm{vol}(\mathcal{M}/G)$ depends on the Riemannian metric, which itself might depend on the group $G$ if we start from a base metric and then deform it to make the action isometric. The volume $\mathrm{vol}(\mathcal{M}_0/G)$ is computed with respect to the dimension of $\mathcal{M}_0/G$, thus being nonzero even if $\dim(\mathcal{M}_0/G) < \dim(\mathcal{M})$.

The effective dimension of the quotient space is defined as $d := \dim(\mathcal{M}_0/G)$. Alternatively, one can define the effective dimension as

$$d := \dim(\mathcal{M}) - \dim(G) + \min_{x \in \mathcal{M}} \dim(G_x), \tag{8}$$

where $G_x := \{\tau \in G : \tau(x) = x\}$ is called the isotropic group of the action at point $x \in \mathcal{M}$. For example, if there exists a point $x \in \mathcal{M}$ with the trivial isotropy group $G_x = \{\mathrm{id}_G\}$, then $d = \dim(\mathcal{M}) - \dim(G)$.

*Remark* 4.2. The invariant Sobolev space satisfies $\mathcal{H}^s_{\mathrm{inv}}(\mathcal{M}) \subseteq \mathcal{H}^{s\theta}_{\mathrm{inv}}(\mathcal{M}) \subseteq L^2_{\mathrm{inv}}(\mathcal{M})$. If the regression function $f^\star$ does not belong to the Sobolev space $\mathcal{H}^s_{\mathrm{inv}}(\mathcal{M})$ (i.e., $\theta \in (0, 1)$), the achieved exponent only depends on $\theta s$ (i.e., the smoothness of the regression function $f^\star$ and not the smoothness of the kernel). The bound decreases monotonically as $s$ increases: smoother functions are easier to learn.

The next theorem states our minimax optimality result. For simplicity, we assume $\theta = 1$.

**Theorem 4.3** (Minimax optimality). *For any estimator $\hat{f}$,*

$$\sup_{\substack{f^\star \in \mathcal{H}^s_{\mathrm{inv}}(\mathcal{M}) \\ \|f^\star\|_{\mathcal{H}^s_{\mathrm{inv}}(\mathcal{M})}=1}} \mathbb{E}\Big[\mathcal{R}(\hat{f}) - \mathcal{R}(f^\star)\Big] \geq C_\kappa \Big(\frac{\omega_d}{(2\pi)^d} \frac{\sigma^2 \, \mathrm{vol}(\mathcal{M}/G)}{n}\Big)^{s/(s+d/2)}, \tag{9}$$

*where $C_\kappa$ is a constant only depending on $\kappa$, and $\omega_d$ is the volume of the unit ball in $\mathbb{R}^d$.*

An explicit formula for $C_\kappa$ is given in the appendix. Note that the above minimax lower bound not only proves that the achieved bound by the KRR estimator is optimal, but also shows the optimality of the prefactor characterized in Theorem 4.1 with respect to the effective dimension $d$ (up to multiplicative constants depending on $\kappa$).

## 4.1 Proof Idea and Dimension Counting Bounds

To prove Theorem 4.1, we develop a general formula for the Fourier series of invariant functions on a manifold. In particular, we argue that a smooth $G$-invariant function $f : \mathcal{M} \to \mathbb{R}$ corresponds to a smooth function $\tilde{f} : \mathcal{M}/G \to \mathbb{R}$ on the quotient space $\mathcal{M}/G$, where $\tilde{f}([x]) = f(x)$ for all $x \in \mathcal{M}$. Hence, we view the space of invariant functions as smooth functions on the quotient space.

The generalization bound essentially depends on a notion of dimension or complexity for this space, which allows bounding the bias and variance terms. We obtain this by controlling the eigenvalues of the Laplace-Beltrami operator, which is specifically suitable for Sobolev kernels.

The *Laplace-Beltrami operator* $\Delta_g$ is the generalization of the usual definition of the Laplacian operator on the Euclidean space $\mathbb{R}^d$ to any smooth manifold. It can be diagonalized in $L^2(\mathcal{M})$ [12]. In particular, there exists an orthonormal basis $\{\phi_\ell(x)\}_{\ell=0}^\infty$ of $L^2(\mathcal{M})$ starting from the constant function $\phi_0 \equiv 1$ such that $\Delta_g \phi_\ell + \lambda_\ell \phi_\ell = 0$, for the discrete spectrum $0 = \lambda_0 < \lambda_1 \leq \lambda_2 \leq \ldots$. Let us call $\{\phi_\ell(x)\}_{\ell=0}^\infty$ the Laplace-Beltrami basis for $L^2(\mathcal{M})$. Our notion of dimension of the function space is $N(\lambda) := \#\{\ell : \lambda_\ell \leq \lambda\}$. It can be shown that if a Lie group $G$ acts smoothly on the compact manifold $\mathcal{M}$, then $G$ also acts on eigenspaces of the Laplace-Beltrami operator. Accordingly, we define the dimension $N(\lambda; G)$ as the dimension of projecting the eigenspaces of the Laplace-Beltrami operator on $\mathcal{M}$ onto the space of $G$-invariant functions, that is, the number of *invariant* eigenfunctions with eigenvalue up to $\lambda$ (Appendix A.7).

Characterizing the asymptotic behavior of $N(\lambda; G)$ is essential for proving our main results on the gain of invariances. Intuitively, the quantity $N(\lambda; G)/N(\lambda)$ corresponds to the fraction of functions that are $G$-invariant. One of this paper's main contributions is to determine the exact asymptotic behavior of this quantity for the analysis of KRR. The tight bound on $N(\lambda; G)$ can be of potential independent interest to other problems related to learning with invariances.

**Theorem 4.4** (Dimension counting theorem). *Let $(\mathcal{M}, g)$ be a smooth connected compact boundaryless Riemannian manifold of dimension $\dim(\mathcal{M})$ and let $G$ be a compact Lie group of dimension $\dim(G)$ acting isometrically on $(\mathcal{M}, g)$. Recall the definition of the effective dimension of the quotient space $d := \dim(\mathcal{M}) - \dim(G) + \min_{x \in \mathcal{M}} \dim(G_x)$. Then,*

$$N(\lambda; G) = \frac{\omega_d}{(2\pi)^d} \operatorname{vol}(\mathcal{M}/G) \lambda^{d/2} + \mathcal{O}(\lambda^{\frac{d-1}{2}}), \tag{10}$$

*as $\lambda \to \infty$, where $\omega_d$ is the volume of the unit ball in $\mathbb{R}^d$.*

In Appendix B, we prove a generalized version of the above bound (i.e., a *local* version).

To see how this counting bound relates to Sobolev spaces, note that by Mercer's theorem, a positive-definite symmetric (PDS) kernel $K : \mathcal{M} \times \mathcal{M} \to \mathbb{R}$ can be diagonalized in an appropriate orthonormal basis of functions in $L^2(\mathcal{M})$. Indeed, the kernel of the Sobolev space $\mathcal{H}^s(\mathcal{M}) \subseteq L^2(\mathcal{M})$ is diagonalizable in the Laplace-Beltrami basis[4]:

$$K_{\mathcal{H}^s(\mathcal{M})}(x, y) = \sum_{\ell=0}^\infty \min(1, \lambda_\ell^{-s}) \phi_\ell(x) \phi_\ell(y), \tag{11}$$

where $\phi_\ell, \ell = 0, 1, \ldots$, form a basis for $L^2(\mathcal{M})$ such that $\Delta_g \phi_\ell + \lambda_\ell \phi_\ell = 0$ for each $\ell$. For the $G$-invariant RKHS $\mathcal{H}_{\mathrm{inv}}^s(\mathcal{M})$, the sum is restricted to $G$-invariant eigenfunctions (Appendix A.9). It is evident that Theorem 4.4 provides an important tool for analyzing the Sobolev kernel development with respect to the eigenfunctions of Laplacian.

## 4.2 Proof Idea of the Dimension Counting Theorem

For Riemannian manifolds, Weyl's law (Appendix A.4) determines the asymptotic distribution of eigenvalues. The bound in Theorem 4.4 indeed corresponds to Weyl's law, if we write it in terms of the quotient space $\mathcal{M}/G$. But, in general, Weyl's law does not apply to the quotient space $\mathcal{M}/G$. So this intuition is not rigorous. First, the quotient space is not always a manifold (Appendix A.5). Second, even if we restrict our attention to the principal part $\mathcal{M}_0/G$ discussed above, which is provably a manifold, other complications arise.

In particular, the quotient $\mathcal{M}_0/G$ can exhibit a boundary, even if the original space is boundaryless. For a concrete example, consider the circle $\mathbb{S}^1 = \{(x, y) : x^2 + y^2 = 1\}$, parameterized by the angle $\theta \in [0, 2\pi)$, under the isometric action $G = \{\mathrm{id}_G, \tau\}$, where $\tau(\theta) = \pi - \theta$ is the reflection across the $y$-axis. The resulting space is a semicircle with two boundary points $(0, \pm 1)$.

---

[4]Many kernels in practice satisfy this condition, e.g., any dot-product kernel on a sphere. While we present the results of this paper for Sobolev kernels, one can use any kernel satisfying the condition in Proposition A.9 and apply the same techniques to obtain its convergence rates.

If the space has a boundary, then a dimension counting result like Weyl's law for manifolds is only true with an appropriate boundary condition for finding the eigenfunctions. In general, different boundary conditions can lead to completely different function spaces for manifolds with boundaries. In the proof, we show that the projections of invariant functions onto the quotient space satisfy the Neumann boundary condition on the (potential) boundaries of the quotient space, thereby exactly characterizing the space of invariant functions, which can indeed be of independent interest.

To see how the Neumann boundary condition appears, consider the circle again and note that its eigenfunctions are the constant function $\phi_0 \equiv 1$, $\sin(k\theta)$, and $\cos(k\theta)$, with eigenvalues $\lambda = k^2$, $k \in \mathbb{N}$. Observe that every eigenvalue is of multiplicity two, except for the zero eigenvalue, which has a multiplicity of one. For the quotient space $\mathbb{S}^1/G$, however, the eigenfunctions are just the constant function, $\sin((2k+1)\theta)$, and $\cos(2k\theta)$, $k \in \mathbb{Z}$ (note how roughly half of the eigenfunctions survived, as $|G| = 2$). In particular, the boundary points $(0, \pm 1)$ satisfy the Neumann boundary condition, while the Dirichlet boundary condition fails to hold; look at the eigenfunctions at $\theta = \pi/2$. More precisely, if we consider the Dirichlet boundary condition, then we get a function space that includes only functions vanishing at the boundary points: $\phi(\pi/2) = \phi(3\pi/2) = 0$. This clearly does not correspond to the space of invariant functions. We generalize this idea in our proof to any manifold and group using differential geometric tools (see Appendix A.5 and Appendix A.6).

In the above example, the boundary points come from the fact that the group action has non-trivial fixed points, i.e., $(0, \pm 1)$ are the fixed points. If the action is free, meaning that we have only trivial fixed points, then the quotient space is indeed a boundaryless manifold. Thus, the challenges towards the proof arise from the existence of non-trivial fixed points.

**Comparison to prior work.** Lastly, we discuss an example on the two-dimensional flat torus $\mathbb{T}^2 = \mathbb{R}^2/2\pi\mathbb{Z}^2$, which shows that the proof ideas in [5] are indeed not applicable for general manifolds even with finite group actions. For this boundaryless manifold, consider the isometric action $G = \{\mathrm{id}_G, \tau\}$, where $\tau(\theta_1, \theta_2) = (\theta_1 + \pi, \theta_2)$, and note that the quotient space $\mathbb{T}^2/G$ is again a torus. In this case, the eigenfunctions of $\mathbb{T}^2$ are the functions $\exp(ik_1 x + ik_2 y)$, for all $k_1, k_2 \in \mathbb{Z}$, with eigenvalue $\lambda = k_1^2 + k_2^2$. But eigenfunctions of the quotient space are those with even $k_1$. This means that some eigenspaces of $\mathbb{T}^2$ (such as those with eigenvalue $\lambda = 2(2n+1)^2$) are completely lost after projection onto the space of invariant functions. This means that the method used in [5] fails to give a non-trivial bound here, because it relies on the fraction of eigenfunctions that survive in each eigenspace. Note that in this example, the quotient space is boundaryless, and the action is free.

## 4.3 Application to Finite-Dimensional Kernels

Applications of Theorem 4.4 are not limited to Sobolev spaces. As an example, we study KRR with finite-dimensional PDS kernels $K : \mathcal{M} \times \mathcal{M} \to \mathbb{R}$ with an RKHS $\mathcal{H} \subseteq L^2(\mathcal{M})$ under invariances (the inclusion must be understood in terms of Hilbert spaces, i.e., the inner product on $\mathcal{H}$ is just the usual inner product defined on $L^2(\mathcal{M})$, making it completely different from Sobolev spaces). Examples of these finite-dimensional spaces are random feature models and two-layer neural networks in the lazy training regime. In this section, we will relate the generalization error of KRR to the average amount of fluctuations of functions in the space and use our dimension counting result to study the gain of invariances.

To formalize this notion of complexity, we need to review some definitions. We measure fluctuation via the *Dirichlet form* $\mathcal{E}$, a bilinear form defined as

$$\mathcal{E}(f_1, f_2) := \int_{\mathcal{M}} \langle \nabla_g f_1(x), \nabla_g f_2(x) \rangle_g d\mathrm{vol}_g(x), \tag{12}$$

for any two smooth functions $f_1, f_2 : \mathcal{M} \to \mathbb{R}$. It can be easily extended (continuously) to any Sobolev space $\mathcal{H}^s(\mathcal{M})$, $s \geq 1$. For each $f \in \mathcal{H}^s(\mathcal{M})$, the diagonal quantity $\mathcal{E}(f, f)$ is called the Dirichlet energy of the function, and is a measure of complexity of a function. Functions with low Dirichlet energy have little fluctuation; intuitively, those have low (normalized) Lipschitz constants on average. Since $\mathcal{E}(af, af) = |a|^2 \mathcal{E}(f, f)$, it is more accurate to restrict to the case $\|f\|_{L^2(\mathcal{M})} = 1$ while studying low-energy functions.

One important example of a finite-dimensional function space is the space of $G$-invariant low-energy functions, which is generated by a finite-dimensional $G$-invariant kernel $K$:

$$K(x, y) = \sum_{\ell=0}^{D-1} \phi_\ell(x)\phi_\ell(y), \tag{13}$$

for any $x, y \in \mathcal{M}$. The non-zero $G$-invariant eigenfunctions $\phi_\ell$ are sorted with respect to their eigenvalues (Appendix A.9). Clearly, $K$ is a kernel of dimension $D$, and it is diagonal in the basis of the Laplace-Beltrami operator's eigenfunctions. The RKHS of $K$ is also of finite dimension $\dim(\mathcal{H}_G) = D$ and can be written as

$$\mathcal{H}_G := \left\{ f \in L^2(\mathcal{M}) : f = \sum_{\ell=0}^{D-1} \langle f, \phi_\ell \rangle_{L^2(\mathcal{M})} \phi_\ell \right\} \subseteq L^2_{\text{inv}}(\mathcal{M}, G). \tag{14}$$

To obtain generalization bounds, we will need to bound a complexity measure of the space of low-energy invariant functions. As a complexity measure for finite function spaces $\mathcal{H} \subseteq L^2_{\text{inv}}(\mathcal{M}, G)$, we use the Dirichlet energy $\mathcal{E}(f, f)$:

$$\mathcal{L}(\mathcal{H}) := \max_{f \in \mathcal{H}} \left\{ \mathcal{E}(f, f) : \|f\|_{L^2(\mathcal{M})} \leq 1 \right\}. \tag{15}$$

For a vector space $\mathcal{H}$, larger $\mathcal{L}(\mathcal{H})$ corresponds to having functions with more (normalized) fluctuation. Thus, $\mathcal{L}(V)$ is a notion of complexity for the vector space $V$. The following proposition shows how $\mathcal{H}_G$ is the *simplest* subspace of $L^2(\mathcal{M})$ with dimension $D$.

**Proposition 4.5.** *For any $D$-dimensional vector space $\mathcal{H} \subseteq L^2_{\text{inv}}(\mathcal{M}, G)$,*

$$\mathcal{L}(\mathcal{H}) \geq \lambda_{D-1}, \tag{16}$$

*and the equality is only achieved when $\mathcal{H} = \mathcal{H}_G$ with $D = \dim(\mathcal{H}_G)$. In particular, if $\mathcal{L}(\mathcal{H}) < \infty$, then $\mathcal{H}$ is of finite dimension. The eigenvalues are sorted according to $G$-invariant eigenspaces (Appendix A.9).*

Using the dimension counting bound in Theorem 4.4, we can explicitly relate the dimension of $\mathcal{H}_G$ to its complexity $\mathcal{L}(\mathcal{H}_G)$. This will be useful for studying the gain of invariances for finite-dimensional kernels.

**Theorem 4.6** (Dimension of the space of low-energy functions)**.** *Under the assumptions in Theorem 4.4, one has the following relation between the dimension of the vector space $\mathcal{H}_G$ and its complexity $\mathcal{L}(\mathcal{H}_G)$:*

$$\dim(\mathcal{H}_G) = \frac{\omega_d}{(2\pi)^d} \operatorname{vol}(\mathcal{M}/G)\mathcal{L}(\mathcal{H}_G)^{d/2} + \mathcal{O}(\mathcal{L}(\mathcal{H}_G)^{\frac{d-1}{2}}), \tag{17}$$

*where $\omega_d$ is the volume of the unit ball in $\mathbb{R}^d$.*

Given the above result, in conjunction with Proposition 4.5, we can obtain the following generalization error for any finite-dimensional RKHS $\mathcal{H} \subseteq L^2(\mathcal{M})$.

**Corollary 4.7** (Convergence rate of KRR with invariances for finite dimensional kernels)**.** *Under the assumptions in Theorem 4.4, for KRR with an arbitrary finite-dimensional $G-$invariant RKHS $\mathcal{H} \subseteq L^2(\mathcal{M})$,*

$$\mathbb{E}\left[\mathcal{R}(\hat{f}) - \mathcal{R}(f^\star)\right] \lesssim \left(\frac{\omega_d}{(2\pi)^d}\frac{\sigma^2 \operatorname{vol}(\mathcal{M}/G)}{n}\right)\mathcal{L}(\mathcal{H})^{d/2}\|f^\star\|^2_{L^2(\mathcal{M})}, \tag{18}$$

*where $\lesssim$ hides absolute constants. Moreover, the upper bound is minimax optimal if $\mathcal{H} = \mathcal{H}_G$ (similar to Theorem 4.3).*

Corollary 4.7 shows the same gain of invariances in terms of sample complexity as Theorem 4.1. Note that in the asymptotic analysis for the above corollary, we assume that $\mathcal{L}(\mathcal{H})$ is large enough, allowing us to use Theorem 4.6.

# 5 Examples and Applications

Next, we make our general results concrete for a number of popular learning settings with invariances. This yields results for kernel versions of popular corresponding architectures.

## 5.1 Sets

When learning with sets, each data instance is a subset $\{x_1, x_2 \ldots, x_m\}$ of elements $x_i \in \mathcal{X}, i \in [m]$, from a given space $\mathcal{X}$. A set is invariant under permutations of its elements, i.e.,

$$\{x_1, x_2 \ldots, x_m\} = \{x_{\sigma_1}, x_{\sigma_2} \ldots, x_{\sigma_m}\}, \tag{19}$$

where $\sigma : [m] \to [m]$ can be any permutation. A successful permutation invariant architecture for learning on sets is DeepSets [70]. Similarly, PointNets are a permutation invariant architecture for point clouds [53, 54]. To analyze learning with sets and kernel versions of these architectures using our formulation, we assume sets of fixed cardinality $m$. If the space $\mathcal{X}$ has a manifold structure, then one can identify each data instance as a point on the product manifold

$$\mathcal{M} = \mathcal{X}^m = \underbrace{\mathcal{X} \times \mathcal{X} \cdots \times \mathcal{X}}_{m}. \tag{20}$$

The task is invariant to the action of the symmetric group $S_m$ on $\mathcal{M}$; each $\sigma \in S_m$ acts on $\mathcal{M}$ by permuting the coordinates as in Equation (19). This action is isometric, and we have $\dim(S_m) = 0$ and $|S_m| = 1/m!$. Theorem 4.1 hence implies that the sample complexity gain from permutation invariance is having effectively $n \times m!$ samples, where $n$ is the number of observed sets. In fact, this result holds (for KRR) for *any* space $\mathcal{X}$ with a manifold structure.

## 5.2 Images

For images, we need translation invariant models. For instance, Convolutional Neural Networks (CNNs) [31, 28] compute translation invariant image representations. Each image is a 2D array $(x_{i,j})_{i,j=0}^{m-1}$ such that $x_{i,j} \in \mathcal{X}$ for a space $\mathcal{X}$ (e.g., for RGB, $\mathcal{X} \subseteq \mathbb{R}^3$ is a compact subset). If $\mathcal{X}$ has a manifold structure, then one can identify each image with a point on the manifold

$$\mathcal{M} = \bigotimes_{i,j=0}^{m-1} \mathcal{X}^{i,j}, \tag{21}$$

where $\mathcal{X}^{i,j}$ is a copy of $\mathcal{X}$. The learning task is invariant under the action of the finite group $(\mathbb{Z}/m\mathbb{Z}) \times (\mathbb{Z}/m\mathbb{Z})$ on $\mathcal{M}$ by shifting pixels: each $(p, q) \in (\mathbb{Z}/m\mathbb{Z}) \times (\mathbb{Z}/m\mathbb{Z})$ corresponds to the isometry $(x_{i,j})_{i,j=1}^m \mapsto (x_{i+p,j+q})_{i,j=1}^m$ (the sum is understood modulo $m$). As a result, the sample complexity gain corresponds to having effectively $n \times m^2$ samples, where $n$ is the number of images.

## 5.3 Point Clouds

3D point clouds have rotation, translation, and permutation symmetries. Tensor field neural networks [62] respect these invariances. We view each 3D point cloud as a set $\{x_1, x_2 \ldots, x_m\}$ such that $x_i \in \mathbb{R}^3/\mathbb{Z}^3 \equiv [0, 1]^3$, which is essentially a point on the manifold $\mathcal{M} = (\mathbb{R}^3/\mathbb{Z}^3)^m$ with $\dim(\mathcal{M}) = 3m$. The learning task is invariant with respect to permuting the coordinates of $\mathcal{M}$, translating all points $x_i \mapsto x_i + r$ for some $r \in \mathbb{R}^3$, and jointly rotating all points, $x_i \mapsto Q x_i$ for an orthogonal matrix $Q$. We denote the group defined by those three operations as $G$ and observe that $\dim(G) = 6$. Thus, the gains of invariances in sample complexity are (1) reducing the dimension $d$ of the space from $3m$ to $3m - 6$, and (2) having effectively $n \times m!$ samples, where $n$ is the number of point clouds.

## 5.4 Sign Flips of Eigenvectors

SignNet [37] is a recent architecture for learning functions of eigenvectors in a spectral decomposition. Each data instance is a sequence of eigenvectors $(v_1, v_2, \ldots, v_m)$, $v_i \in \mathbb{R}^d$, and flipping the sign of an eigenvector $v_i \to -v_i$ does not change its eigenspace. The spectral data can be considered as a point on the manifold $\mathcal{M} = (\mathbb{S}^{d-1})^m$ (where $\mathbb{S}^{d-1}$ is the $(d-1)$-dimensional sphere), while the task is invariant to all $2^m$ possibilities of sign flips. The sample complexity gain of invariances is thus having effectively $n \times 2^m$ samples, where $n$ is the number of spectral data points.

## 5.5 Changes of Basis for Eigenvectors

BasisNet [37] represents spectral data with eigenvalue multiplicities. Each input instance is a sequence of eigenspaces $(V_1, V_2, \ldots, V_p)$, and each $V_i$ is represented by an orthonormal basis such

as $(v_{i,1}, v_{i,2}, \ldots, v_{i,m_i}) \in (\mathbb{R}^d)^{m_i}$. This is the *Stiefel manifold* with dimension $dm_i - \frac{m_i(m_i+1)}{2}$. Thus, the spectral data lie on a manifold of dimension

$$\dim(\mathcal{M}) = \sum_{i=1}^{p} \left( dm_i - m_i(m_i + 1)/2 \right). \tag{22}$$

The vector spaces' representations are invariant to a change of basis, i.e., the group action is defined as $(v_{i,1}, v_{i,2}, \ldots, v_{i,m_i}) \mapsto (Qv_{i,1}, Qv_{i,2}, \ldots, Qv_{i,m_i})$ for any orthogonal matrix $Q$ that fixes the eigenspace $V_i$. If $G$ denotes this group of invariances, then

$$\dim(G) = \sum_{i=1}^{p} \left( m_i(m_i - 1)/2 \right). \tag{23}$$

Thus, the gain of invariances is a reduction of the manifold's dimension to $\sum_{i=1}^{p}(dm_i - m_i^2)$. For example, if $m_i = m$ for all $i$, then with $d = pm$ we get $\dim(\mathcal{M}) = d^2 - \frac{1}{2}d(m + 1)$ while after the reduction we have $\dim(\mathcal{M}/G) = d^2 - dm$. In this example, the quotient space is the *Grassmannian manifold*.

## 5.6 Learning on Graphs

Each weighted, possibly directed graph on $m$ vertices with initial node attributes can be naturally encoded by its adjacency matrix $A \in \mathbb{R}^{m \times m}$. Thus, the space of all weighted directed graphs on $m$ vertices corresponds to a compact manifold $\mathcal{M} \subseteq \mathbb{R}^{m \times m}$ if the weights are restricted to a bounded set. Learning tasks on graphs are invariant to permutations of rows and columns, i.e., the action of the symmetric group as $A \mapsto P^{-1}AP$ for any permutation matrix $P$. For instance, Graph Neural Networks (GNNs) and graph kernels [64] implement this invariance. The sample complexity gain from invariances is thus evident; it corresponds to having effectively $n \times m!$ samples, where $n$ is the number of sampled graphs.

## 5.7 Hyperbolic Spaces, Tori, etc.

One important feature of the results in this paper is that they are not restricted to compact submanifolds of Euclidean spaces. In particular, the results are also valid for compact hyperbolic spaces; see [51] for a survey on applications of hyperbolic spaces in machine learning. Another type of space where our results are still valid are tori, i.e., $\mathbb{T}^d := (\mathbb{S}^1)^d$. Tori naturally occur for modeling joints in robot control [38]. In fact, Riemannian manifolds are beneficial in a broader context for learning in robotics, e.g., arising from constraints, as surveyed in [10]. Our results apply to invariant learning in all of these settings, too.

# 6 Conclusion

In this work, we derived new generalization bounds for learning with invariances. These generalization bounds show a two-fold gain in sample complexity: (1) in the dimension term in the exponent, and (2) in the effective number of samples. Our results significantly generalize the range of settings where the bounds apply. In particular, (1) goes beyond prior work, since it applies to groups of positive dimension, whereas prior work assumed finite dimensions. At the heart of our analysis is a new dimension counting bound for invariant functions on manifolds, which we expect to be useful more generally for analyzing learning with invariances. We prove this bound via differential geometry and show how to overcome several technical challenges related to the properties of the quotient space.

## Acknowledgments and Disclosure of Funding

The authors extend their appreciation to Semyon Dyatlov and Henrik Christensen for their valuable recommendations and inspiration. This work was supported by Office of Naval Research grant N00014-20-1-2023 (MURI ML-SCOPE), NSF award CCF-2112665 (TILOS AI Institute), NSF award 2134108, and NSF TRIPODS program (award DMS-2022448).

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

# A  Preliminaries

## A.1  Manifolds

A completely metrizable second countable topological space $\mathcal{M}$ that is locally homeomorphic to the Euclidean space $\mathbb{R}^{\dim(\mathcal{M})}$ is called a manifold of dimension $\dim(\mathcal{M})$. Sphere, torus, and $\mathbb{R}^d$ are some examples of manifolds. A Riemannian manifold $(\mathcal{M}, g)$ is a manifold that is equipped with a smooth inner product $g$ on the tangent space $T_x\mathcal{M}$ around each point $x \in \mathcal{M}$. A Riemannian metric tensor $g$ allows the use of the following two important tools in the study of manifolds: (i) the geodesic distance $d_{\mathcal{M}}(x, y)$ between any points $x, y \in \mathcal{M}$, and (ii) the volume element $d\,\mathrm{vol}_g(x)$, that defines a Borel measure on the manifold (on the Borel sigma-algebra of open subsets of the manifold). The distance function and the volume element depend on the Riemannian metric tensor $g$ and make the manifold a metric-measure space. The functional spaces on manifolds, such as $L^p(\mathcal{M})$, $p \in [1, \infty]$, and the Sobolev spaces $\mathcal{H}^s(\mathcal{M})$, $s \geq 0$, are defined similar to the Euclidean spaces.

## A.2  Isometries

A bijection $\tau : \mathcal{M} \to \mathcal{M}$ is called an isometry of $(\mathcal{M}, g)$, if and only if $d_{\mathcal{M}}(\tau(x), \tau(y)) = d_{\mathcal{M}}(x, y)$ for all $x, y \in \mathcal{M}$. The space of isometries of $(\mathcal{M}, g)$ forms a group under composition, denoted by $\mathrm{ISO}(\mathcal{M}, g)$ or simply by $\mathrm{ISO}(\mathcal{M})$. By the Myers–Steenrod theorem, for connected manifolds, each $\tau \in \mathrm{ISO}(M, g)$ is not only bijective but indeed smooth (i.e., infinitely many times differentiable) [52]. Moreover, $\mathrm{ISO}(\mathcal{M}, g)$ is a Lie group (i.e., a smooth manifold which is simultaneously a group) of dimension at most $\frac{\dim(\mathcal{M})(\dim(\mathcal{M})+1)}{2}$. Another characterization of isometries on manifolds is based on metric tensors, where a function $\tau : \mathcal{M} \to \mathcal{M}$ is an isometry if and only if the pullback of the metric tensor $g$ by $\tau$ is itself, i.e., $g = \tau^* g$. This means that $\tau$ locally preserved inner products of tangent vectors. For the applications in this paper, whenever it is not mentioned, a $\dim(\mathcal{M})-$dimensional Riemannian manifolds $(\mathcal{M}, g)$ is smooth, connected, and compact boundaryless.

## A.3  Laplacain on Manifolds

The *Laplace-Beltrami operator* is the unique continuous linear operator $\Delta_g : \mathcal{H}^s(\mathcal{M}) \to \mathcal{H}^{(s-2)}(\mathcal{M})$ satisfying the property

$$\int_{\mathcal{M}} \psi(x) \Delta_g \phi(x) d\mathrm{vol}_g(x) = -\int_{\mathcal{M}} \langle \nabla_g \phi(x), \nabla_g \psi(x) \rangle_g d\mathrm{vol}_g(x), \qquad (24)$$

for any $\phi, \psi \in \mathcal{H}^s(\mathcal{M})$. This generalizes the usual definition of the Laplacian operator $\Delta = \partial_1^2 + \cdots + \partial_d^2$, defined on the Euclidean space $\mathbb{R}^d$, which satisfies this property by integration by parts. The kernel of the operator $\Delta_g$ includes the so-called harmonic functions. In the case of compact boundaryless manifolds, the only harmonic functions are constants.

The operator $(-\Delta_g)$ is elliptic, self-adjoint, and can be diagonalized in $L^2(\mathcal{M})$ [12]. In particular, there exists an orthonormal basis $\{\phi_\ell(x)\}_{\ell=0}^{\infty}$ of $L^2(\mathcal{M})$ starting from the constant function $\phi_0 \equiv 1$ such that $\Delta_g \phi_\ell + \lambda_\ell \phi_\ell = 0$, for the discrete spectrum $0 = \lambda_0 < \lambda_1 \leq \lambda_2 \leq \ldots$. Note that eigenvalues appear in this sequence with their multiplicities. Let us call $\{\phi_\ell(x)\}_{\ell=0}^{\infty}$ the Laplace-Beltrami basis for $L^2(\mathcal{M})$. Using the above identity, for an arbitrary $f \in L^2(\mathcal{M})$ with the expansion $f = \sum_{\ell=0}^{\infty} \langle f, \phi_\ell \rangle_{L^2(\mathcal{M})} \phi_\ell$ and $L^2(\mathcal{M})-$norm $\quad \|f\|_{L^2(\mathcal{M})}^2 = \sum_{\ell=0}^{\infty} |\langle f, \phi_\ell \rangle_{L^2(\mathcal{M})}|^2$, we have $\Delta_g f = -\sum_{\ell=0}^{\infty} \lambda_\ell \langle f, \phi_\ell \rangle \phi_\ell$ and

$$\|\nabla_g f\|_{L^2(\mathcal{M})}^2 = \int_{\mathcal{M}} \langle \nabla_g f(x), \nabla_g f(x) \rangle_g d\mathrm{vol}_g(x) \qquad (25)$$

$$= -\int_{\mathcal{M}} f(x) \Delta_g f(x) d\mathrm{vol}_g(x) \qquad (26)$$

$$= \sum_{\ell=0}^{\infty} \lambda_\ell \langle f, \phi_\ell \rangle_{L^2(\mathcal{M})}^2, \qquad (27)$$

whenever the sums converge.

## A.4 (Local) Weyl's Law

It is known that the Laplace-Beltrami spectrum can encode many manifold geometric properties, such as dimension, volume, etc. However, it is also known that isospectral (non-isomorphic) manifolds exist. Still, it provides rich information about the manifold. Let us denote the set of distinct eigenvalues of a manifold by $\text{Spec}(\mathcal{M}) := \{\lambda_0, \lambda_1, \ldots\} \subset \mathbb{R}_{\geq 0}$.

Weyl's law characterizes the asymptotic distribution of the eigenvalues in a closed-form formula. Let us denote the number of eigenvalues of the Laplace-Beltrami operator up to $\lambda$ as $N(\lambda) := \#\{\ell : \lambda_\ell \leq \lambda\}$.

**Proposition A.1** (Local Weyl's law, [11, 25, 60]). *Let $(\mathcal{M}, g)$ denote an arbitrary* $\dim(\mathcal{M})-$*dimensional compact boundaryless Riemannian manifold. Then, for all $x \in \mathcal{M}$,*

$$N_x(\lambda) := \sum_{\ell:\lambda_\ell \leq \lambda} |\phi_\ell(x)|^2 = \frac{\omega_{\dim(\mathcal{M})}}{(2\pi)^{\dim(\mathcal{M})}} \text{vol}(\mathcal{M})\lambda^{\dim(\mathcal{M})/2} + \mathcal{O}(\lambda^{\frac{\dim(\mathcal{M})-1}{2}}), \qquad (28)$$

*as $\lambda \to \infty$, where $\omega_d = \frac{\pi^{d/2}}{\Gamma(\frac{d}{2}+1)}$ is the volume of the unit $d-$ball in the Euclidean space $\mathbb{R}^d$. The constant in the error term is indeed independent of $x$ and may only depend on the sectional curvature and the injectivity radius of the manifold [16]. As a byproduct, the following $L^\infty$ upper bound also holds:*

$$\max_{\lambda_\ell \leq \lambda} \max_{x \in \mathcal{M}} |\phi_\ell(x)| = \mathcal{O}\left(\lambda^{\frac{\dim(\mathcal{M})-1}{4}}\right). \qquad (29)$$

By definition, $N(\lambda) = \int_\mathcal{M} N_x(\lambda)d\,\text{vol}_g(x)$ and thus, while the above result is called the local Weyl's law, the traditional Weyl's law can be easily derived as $N(\lambda) = \frac{\omega_{\dim(\mathcal{M})}}{(2\pi)^{\dim(\mathcal{M})}} \text{vol}(\mathcal{M})\lambda^{\dim(\mathcal{M})/2} + \mathcal{O}(\lambda^{\frac{\dim(\mathcal{M})-1}{2}})$. For compact Riemannian manifolds $(\mathcal{M}, g)$ with boundary, to define the eigenfunctions of the Laplace-Beltrami operator (i.e., the solutions to the equation $\Delta_g\phi + \lambda\phi = 0$), one has to consider boundary conditions. For the Dirichlet boundary condition (i.e., assuming the solution vanishes on the boundary), or the Neumann boundary condition (i.e., assuming the solution's gradient vanishes on the outward normal vector at each point of the boundary), the above claim on the local behavior of eigenfunctions still holds.

The asymptotic distribution of eigenvalues allows us to define the Minakshisundaram–Pleijel zeta function of the manifold $(\mathcal{M}, g)$ as follows [47]:

$$\mathcal{Z}_\mathcal{M}(s) = \sum_{\ell=1}^\infty \lambda_\ell^{-s} = \int_{0+}^\infty \lambda^{-s}dN(\lambda); \quad \Im(s) > \frac{\dim(\mathcal{M})}{2}, \qquad (30)$$

where the sum converges absolutely by Weyl's law. Note that the integral must be understood as a Riemann–Stieltjes integral. The zeta function can be analytically continued to a meromorphic function on the complex plane and has a functional equation. By integration by parts,

$$\mathcal{Z}_\mathcal{M}(s) = \int_{0+}^\infty \lambda^{-s}dN(\lambda) \qquad (31)$$

$$= \lambda^{-s}N(\lambda)\big|_{0+}^\infty - \int_{0+}^\infty N(\lambda)(-s)\lambda^{-s-1}d\lambda \qquad (32)$$

$$= s\int_{0+}^\infty N(\lambda)\lambda^{-s-1}d\lambda, \qquad (33)$$

by $N(\lambda) = \mathcal{O}(\lambda^{\dim(\mathcal{M})/2})$ and the assumption $\Im(s) > \frac{\dim(\mathcal{M})}{2}$.

For more information on the spectral theory of Riemannian manifolds, see [29].

## A.5 Quotient Manifold Theorem

This part and the next subsection review some classical results about group actions on manifolds (mostly from [33, 27, 7]). Let $G$ be an arbitrary group. The action of $G$ on the manifold $\mathcal{M}$ is a mapping $\theta : G \times \mathcal{M} \to \mathcal{M}$ such that $\theta(\text{id}_G, .) = \text{id}_\mathcal{M}$ and $\theta(\tau_1\tau_2, .) = \theta(\tau_1, \theta(\tau_2, .))$ for any

$\tau_1, \tau_2 \in G$. In particular, any group action gives a $G-$indexed set of bijections on $\mathcal{M}$ with respect to the group law of $G$. For example, $\text{ISO}(\mathcal{M})$ acts on $\mathcal{M}$ by the isometric transformations (which are bijections by definition).

For each $x \in \mathcal{M}$, the orbit of $x$ is defined as the set of all images of the group transformations at $x$ on the manifold:

$$[x] := \big\{\theta(\tau, x) \in \mathcal{M} : \tau \in G\big\}. \tag{34}$$

Also, for any $x \in \mathcal{M}$, define the isotropy group $G_x := \{\tau \in G : \theta(\tau, x) = x\}$. In other words, the isotropy group $G_x$, as a subgroup of $G$, includes all the transformations $\tau \in G$ which have $x \in \mathcal{M}$ as a fixed point. The set of all orbits is denoted by $\mathcal{M}/G$ and is called the orbit space (or sometimes the fundamental domain) of the action on the manifold:

$$\mathcal{M}/G := \big\{[x] : x \in \mathcal{M}\big\}. \tag{35}$$

It is known that $\mathcal{M}/G$ admits a unique topological structure coming from the topology of the manifold, making the projection (or the quotient) map $\pi : \mathcal{M} \to \mathcal{M}/G$ continuous.

However, $\mathcal{M}/G$ is not always a manifold. For example, if $\mathcal{M} = \mathbb{R}^d$ and $G = \text{GL}_d(\mathbb{R})$, then the resulting orbit space $\mathcal{M}/G$ is not Hausdorff. Even in cases where it is a manifold, the orbit space $\mathcal{M}/G$ may have boundaries while $\mathcal{M}$ is boundaryless. For example, consider the action of the orthogonal group $\mathcal{O}(d) := \{A \in \text{GL}_d(\mathbb{R}) : A^T A = A^T A = I_d\}$ on the manifold $\mathcal{M} = \mathbb{R}^d$, where the orbit space becomes $\mathcal{M}/G = \mathbb{R}_{\geq 0}$. For the purpose of this paper, boundaries are important and affect the results drastically.

The quotient manifold theorem gives a number of sufficient conditions on the manifold/group, such that the orbit space is always a manifold. To introduce the theorem, we need to review a few classical definitions as follows.

A group action is called free, if and only if it has no non-trivial fixed points, i.e., $\theta(\tau, x) \neq x$ for all $\tau \neq \text{id}_G$ (equivalently, $G_x = \{\text{id}_G\}$ for all $x \in \mathcal{M}$). For example, the action of the group of linear transformations $\theta(r, x) = x + r$, for each $x, r \in \mathbb{R}^d$, on the manifold $\mathbb{R}^d$ is a free action. An action is called smooth if and only if for each $\tau \in G$, the mapping $\theta(\tau, .) : \mathcal{M} \to \mathcal{M}$ is smooth. An action is called proper, if and only if the map $\big(\theta(\tau, x), x\big) : G \times \mathcal{M} \to \mathcal{M} \times \mathcal{M}$ is a proper map. As a sufficient condition, every continuous action of a compact Lie group $G$ on a manifold is proper. To simplify our notation, let us define $\tau(x) := \theta(\tau, x)$.

**Theorem A.2** (Quotient Manifold Theorem, [33])**.** *Let $G$ be a Lie group acting smoothly, freely, and properly on a smooth manifold $(\mathcal{M}, g)$. Then, the orbit space (or the fundamental domain) $\mathcal{M}/G$ is a smooth manifold of dimension $\dim(\mathcal{M}) - \dim(G)$ with a unique smooth structure such that the projection (or the quotient) map $\pi : \mathcal{M} \to \mathcal{M}/G$ is a smooth submersion.*

**Corollary A.3.** *Suppose $\mathcal{M}$ is a connected smooth compact boundaryless manifold. Then, the isometry group $\text{ISO}(\mathcal{M})$ is a compact Lie group acting smoothly on $\mathcal{M}$. Thus, if $G$ is a closed subgroup of $\text{ISO}(\mathcal{M})$, then the action of $G$ on $\mathcal{M}$ is smooth and proper. In addition, if this action is free, then the orbit space $\mathcal{M}/G$ becomes a connected closed (i.e, compact boundaryless) manifold of dimension $\dim(\mathcal{M}) - \dim(G)$.*

*Furthermore, assuming $(\mathcal{M}, g)$ is a Riemannian manifold, there exists a unique Riemannian metric $\tilde{g}$ such that the projection map $\pi : (\mathcal{M}, g) \to (\mathcal{M}/G, \tilde{g})$ is a Riemannian submersion.*

There is indeed a natural way to interpret these results. Given a manifold $(\mathcal{M}, g)$, the orbit space is somehow the shrinkage of the manifold to represent exactly one representative from each orbit, and the metric $\tilde{g}$ is just identical to the original metric if the tangent lines survive; otherwise, the tangent lines are killed, and the inner product defined by $\tilde{g}$ is zero.

## A.6  Principal Orbit Theorem

The quotient manifold theorem is, however, restricted to free actions. Unfortunately, this assumption is generally necessary to prove that the quotient space is a compact boundaryless manifold. In the lack of freeness, it is still possible to show that the quotient space is *almost* a manifold (possibly with boundary). First, let us introduce a few definitions. Two isotropy groups $G_x$ and $G_y$ are called conjugate (with respect to $G$), if and only if $\tau^{-1} G_x \tau = G_y$ for some $\tau \in G$. This is a relation on the

manifold, and the *isotropy type* of any $x \in \mathcal{M}$ is defined as the equivalence class of its isotropy group, denoted as $[G_x]$. Since all the points on an orbit have the same isotropy type (by definition), one can also define the isotropy type of an orbit in the same way. Given the conjugacy relation, a partial order can be naturally defined as follows: $H_1 \preceq H_2$ if and only if $H_1$ is conjugate to a subgroup of $H_2$. Since the definition is unchanged modulo conjugate groups, one can also restrict the partial order to the conjugacy class of subgroups. This allows us to define a partial order on orbits as well: for any two orbits $[x], [y]$, we write $[x] \leq [y]$ if and only if $G_y \preceq G_x$. For example, if $G_x = \{\mathrm{id}_G\}$, then $[y] \leq [x]$ for all $y \in \mathcal{M}$.

Given the above formal definitions, the quotient manifold theorem assumes that *all* the orbits have a *unique maximal orbit type*, namely, the orbit type $[\{\mathrm{id}_G\}]$. By removing the freeness assumption, however, there might be several orbit types that are not necessarily comparable with respect to the partial order. However, the *principal orbit type theorem* shows that there always exists a *unique maximal orbit type*, where when the action is restricted to those orbits, it defines a Riemannian submersion (thus the image is a manifold), and moreover, the *principal orbits* (those orbit with the maximal orbit types) are *dense* in the manifold $\mathcal{M}$. This shows that we almost get a nice space, which suffices for our subsequent proofs in the next sections.

**Theorem A.4** (Principal Orbit Theorem). *Let $G$ be a compact Lie group acting isometrically on a Riemannian manifold $(\mathcal{M}, g)$. Then, there exists an open dense subset $\mathcal{M}_0 \subseteq \mathcal{M}$, such that $[x] \geq [y]$ for all $x \in \mathcal{M}_0$ and $y \in \mathcal{M}$. Moreover, the natural projection $\pi : \mathcal{M}_0 \to \mathcal{M}_0/G \subseteq \mathcal{M}/G$ is a Riemannian submersion. Also, the set $\mathcal{M}_0/G \subseteq \mathcal{M}/G$ is open, dense, and connected in $\mathcal{M}/G$.*

**Corollary A.5.** *One has the decomposition*

$$\mathcal{M}/G = \bigsqcup_{[H] \preceq G} \mathcal{M}_{[H]}/G, \tag{36}$$

*where $\mathcal{M}_{[H]} := \{x \in \mathcal{M} : [G_x] = [H]\}$ is a submanifold of $\mathcal{M}$. The disjoint union is taken over all isotropy types of the group action on the manifold. The map $\pi : \mathcal{M}_{[H]} \to \mathcal{M}_{[H]}/G$ is a Riemannian submersion; therefore its image is a manifold. By an application of the Slice theorem, one can observe that only finitely many isotropy types can exist when $G$ and $\mathcal{M}$ are both compact. Thus, the disjoint union is indeed over finitely many precompact smooth manifolds. Among those, the principal part $\mathcal{M}_0/G := \mathcal{M}_{[H_0]}/G$ is dense in $\mathcal{M}/G$, where $[H_0]$ is the unique maximal isotropy type of the group action.*

Intuitively, the above corollary shows that the quotient space can be decomposed to finitely many "pieces," and each piece has a nice smooth structure. In the case of a free action, the decomposition above reduces to just one "piece" with the unique trivial maximal orbit type (i.e., having a trivial isotropy group). The dimension of each "piece" $\mathcal{M}_{[H]}/G$ can be computed as

$$\dim(\mathcal{M}_{[H]}/G) = \dim(\mathcal{M}) - \dim(G) + \dim(H). \tag{37}$$

The *effective dimension of the quotient space* is then defined as

$$d := \dim(\mathcal{M}_{[H_0]}/G) = \dim(\mathcal{M}_0/G) = \dim(\mathcal{M}) - \dim(G) + \dim(H_0), \tag{38}$$

where $[H_0]$ is the unique maximal isotropy type of the group action.

## A.7 Isometries and Laplace-Beltrami Eigenspaces

In Weyl's law, the eigenvalues are counted with their multiplicities. Let us define $V_\lambda$ as the eigenspace of the eigenvalue $\lambda$ with the finite dimension $\dim(V_\lambda)$. Then, $N(\lambda) = \sum_{\lambda' \leq \lambda} \dim(V_{\lambda'})$.

As a well-known fact from differential geometry, the Laplace-Beltrami operator $\Delta_g$ commutes with all isometries $\tau \in \mathrm{ISO}(\mathcal{M})$. Thus,

$$\Delta_g \phi_\ell + \lambda_\ell \phi_\ell = 0 \iff \Delta_g(\phi_\ell \circ \tau) + \lambda_\ell(\phi_\ell \circ \tau) = 0. \tag{39}$$

This means that the eigenspaces of the Laplace-Beltrami operator are invariant with respect to the action of the isometry group on the manifold.

**Corollary A.6.** *$L^2(\mathcal{M}) = \bigoplus_{\lambda \in \mathrm{Spec}(\mathcal{M})} V_\lambda$, and the isometry group of the manifold $\mathrm{ISO}(\mathcal{M})$ (and thus all its closed subgroups) acts on each eigenspace $V_\lambda$.*

Now consider an arbitrary closed subgroup $G$ of the isometry group $\mathrm{ISO}(\mathcal{M})$. Then $G$ acts on the eigenspaces of the Laplace-Beltrami operator and each $\tau \in G$ corresponds to a bijective linear transformation $f \mapsto f \circ \tau$, denoted as $\tau_\lambda^* : V_\lambda \to V_\lambda$. There is a natural way to extend this operator into the whole space $L^2(\mathcal{M})$ so one may consider $\tau_\lambda^* : L^2(\mathcal{M}) \to L^2(\mathcal{M})$. Since $\tau$ is an isometry, for any $\phi \in V_\lambda$,

$$\|\tau_\lambda^* \phi\|_{L^2(\mathcal{M})}^2 = \int_\mathcal{M} |\tau_\lambda^* \phi(x)|^2 d\mathrm{vol}_g(x) \tag{40}$$

$$= \int_\mathcal{M} |\phi(x)|^2 d\mathrm{vol}_g(x) \tag{41}$$

$$= \|\phi\|_{L^2(\mathcal{M})}^2 \tag{42}$$

$$= 1, \tag{43}$$

since $d\,\mathrm{vol}_g$ is invariant with respect to any isometry $\tau \in G$. This shows that the bijective linear transformation $\tau_\lambda^*$ is indeed a representation of the group $G$ into the orthogonal group $\mathcal{O}(\dim(V_\lambda))$ for each eigenvalue $\lambda$.

In this paper, we are interested in the space of invariant functions defined with respect to an arbitrary closed subgroup $G$ of the isometry group $\mathrm{ISO}(\mathcal{M})$.

**Definition A.7.** The space of invariant functions with respect to a closed subgroup $G$ of the isometry group $\mathrm{ISO}(\mathcal{M})$ is defined as

$$L_{\mathrm{inv}}^2(\mathcal{M}, G) := \left\{ f \in L^2(\mathcal{M}) : \forall \tau \in G : \tau^* f := f \circ \tau = f \right\} \subseteq L^2(\mathcal{M}), \tag{44}$$

as a closed subspace of $L^2(\mathcal{M})$.

Let $d\tau$ denote the Haar measure (i.e., the uniform measure) associated with a closed subgroup $G$ of the isometry group $\mathrm{ISO}(\mathcal{M})$. Let the projection operator $\mathcal{P}_G : L^2(\mathcal{M}) \to L_{\mathrm{inv}}^2(\mathcal{M}, G)$ be defined as $f(x) \mapsto \int_G (f \circ \tau)(x) d\tau = \int_G \tau^* f(x) d\tau$. Claerly, $f \in L_{\mathrm{inv}}^2(\mathcal{M}, G)$ if and only if $f \in \ker(I - \mathcal{P}_G)$.

**Proposition A.8.** *For any closed subgroup $G$ of the isometry group $\mathrm{ISO}(\mathcal{M})$, the following decomposition holds:*

$$L_{\mathrm{inv}}^2(\mathcal{M}, G) = \ker(I - \mathcal{P}_G) \tag{45}$$

$$= \bigcap_{\substack{\lambda \in \mathrm{Spec}(\mathcal{M}) \\ \tau \in G}} \ker(I - \tau_\lambda^*) \tag{46}$$

$$= \bigoplus_{\lambda \in \mathrm{Spec}(\mathcal{M})} V_{\lambda,G}, \tag{47}$$

*where each $V_{\lambda,G}$ is a linear subspace of $V_\lambda$ defined as*

$$V_{\lambda,G} := \left\{ f \in V_\lambda : \forall \tau \in G : \tau_\lambda^* f := f \circ \tau = f \right\} \tag{48}$$

$$= \bigcap_{\tau \in G} \ker(I - \tau_\lambda^*). \tag{49}$$

*Clearly, $\dim(V_{\lambda,G}) \leq \dim(V_\lambda)$.*

*Moreover, the (restricted) projection operator $\mathcal{P}_G : V_\lambda \to V_\lambda$ has the image $V_{\lambda,G}$ and it can be diagonalized in a basis for $V_\lambda$ such as $\phi_{\lambda,\ell} \in V_\lambda, \ell = 1, 2, \ldots, \dim(V_\lambda)$, such that for each $f = \sum_{\ell=1}^{\dim(V_\lambda)} \alpha_\ell \phi_{\lambda,\ell} \in V_\lambda$,*

$$f \in V_{\lambda,G} \iff \forall \ell > \dim(V_{\lambda,G}) : \alpha_\ell = 0 \tag{50}$$

$$\mathcal{P}_{\lambda,G} f = \sum_{\ell=1}^{\dim(V_{\lambda,G})} \alpha_\ell \phi_{\lambda,\ell} \in V_{\lambda,G}. \tag{51}$$

Due to its simplicity, we omit the proof of this proposition. We always consider the diagonalized basis in this paper, as it always exists for appropriate eigenfunctions.

## A.8 Reproducing Kernel Hilbert Spaces on Manifolds

A smooth connected compact boundaryless Riemannian manifold $(\mathcal{M}, g)$ is indeed a compact metric-measure space, and a kernel $K : \mathcal{M} \times \mathcal{M} \to \mathbb{R}$ can be thought of as a measure of similarity between points on the manifold. We assume that $K$ is continuous, symmetric, and positive-definite, meaning that $K(x, y) = K(y, x)$ and $\sum_{i,j=1}^{n} a_i a_j K(x_i, y_j) \geq 0$ for any $a_i \in \mathbb{R}, x_i, y_j \in \mathcal{M}, i, j = 1, 2, \ldots, n$, and the equality happens only when $a_1 = a_2 = \ldots = a_n = 0$ (assuming the points on the manifold are distinct). The Reproducing Kernel Hilbert Space (RKHS) of $K$ is a Hilbert space $\mathcal{H} \subseteq L^2(\mathcal{M})$ that is achieved by the completion of the span of functions $K(., y) \in \mathcal{H}$ for each $y \in \mathcal{M}$, satisfying the following property: for all $f \in \mathcal{H}$, $f(x) = \langle f, K(x,.) \rangle_{\mathcal{H}}$. Associated with the kernel $K$, there exists an integral operator $\mathcal{K} : L^2(\mathcal{M}) \to L^2(\mathcal{M})$ defined as $\mathcal{K}(f) = \int_{\mathcal{M}} K(x, y) f(y) d \operatorname{vol}_g(y)$. It can be shown that $K$ can be diagonalized in an appropriate orthonormal basis of functions in $L^2(\mathcal{M})$ (Mercer's theorem). Indeed, with a number of appropriate assumptions, it can be diagonalized in the Laplace-Beltrami spectrum.

**Proposition A.9.** *Consider a symmetric positive-definite kernel $K : \mathcal{M} \times \mathcal{M} \to \mathbb{R}$, and assume that $K \in C^2(\mathcal{M} \times \mathcal{M})$ satisfies the differential equation $\Delta_{g,x}(K(x, y)) = \Delta_{g,y}(K(x, y))$. Then, $K$ can be diagonalized in the basis of the eigenfunctions of the Laplace-Beltrami operator; there exist appropriate $\mu_\ell \geq 0$, $\ell = 0, 1, \ldots$, such that*

$$K(x, y) = \sum_{\ell=0}^{\infty} \mu_\ell \phi_\ell(x) \phi_\ell(y), \tag{52}$$

*where $\phi_\ell, \ell = 0, 1, \ldots$, form an orthonormal basis for $L^2(\mathcal{M})$ such that $\Delta_g \phi_\ell + \lambda_\ell \phi_\ell = 0$ for each $\ell$.*

*Proof.* Note that $\mathcal{K}(\phi_\ell(x)) = \int_{\mathcal{M}} K(x, y) \phi_\ell(y) d \operatorname{vol}_g(y)$. Therefore,

$$\Delta_{g,x}(\mathcal{K}(\phi_\ell)) = \Delta_{g,x}\left( \int_{\mathcal{M}} K(x, y) \phi_\ell(y) dvol_g(y) \right) \tag{53}$$

$$= \int_{\mathcal{M}} \Delta_{g,x} K(x, y) \phi_\ell(y) dvol_g(y) \tag{54}$$

$$= \int_{\mathcal{M}} \Delta_{g,y} K(x, y) \phi_\ell(y) dvol_g(y) \tag{55}$$

$$= \int_{\mathcal{M}} K(x, y) \Delta_{g,y} \phi_\ell(y) dvol_g(y) \tag{56}$$

$$= \lambda_\ell \int_{\mathcal{M}} K(x, y) \phi_\ell(y) dvol_g(y) \tag{57}$$

$$= \lambda_\ell \mathcal{K}(\phi_\ell), \tag{58}$$

where we used the symmetry of the kernel, the regularity condition of the kernel (allowing the interchange of the differentiation and the integral sign), and also the self-adjointness of the Laplace-Beltrami operator. Now since $\mathcal{K}(\phi_\ell)$ satisfies the equation $\Delta_g(\mathcal{K}(\phi_\ell)) + \lambda_i \mathcal{K}(\phi_\ell) = 0$, we conclude that $\mathcal{K}(\phi_\ell)$ is indeed an eigenfunction with respect to the eigenvalue $\lambda_\ell$, or equivalently $\mathcal{K}(\phi_\ell) \in V_{\lambda_\ell}$. In other words, $V_\lambda$, $\lambda \in \operatorname{Spec}(\mathcal{M})$, are the invariant subspaces of the integral operator associated with the kernel. This means that one can choose an appropriate basis of eigenfunctions in each eigenspace, such that the kernel is diagonalized in each eigenspace (Mercer's theorem). $\qquad \square$

*Remark* A.10. While Proposition A.9 holds for a kernel $K \in C^2(\mathcal{M} \times \mathcal{M})$, it can be shown that it holds under a weaker assumption that $K$ is just continuous. The identity $\Delta_{g,x}(K(x, y)) = \Delta_{g,y}(K(x, y))$ should then be understood as the identity of two distributions.

In this paper, we always consider the diagonalized kernels in the Laplace-Beltrami spectrum. An example of a kernel of this form is the heat kernel with $\mu_\ell = e^{-\lambda_\ell t}$, $t \in \mathbb{R}$. Given a diagonalized kernel $K(x, y) = \sum_{\ell=0}^{\infty} \mu_\ell \phi_\ell(x) \phi_\ell(y)$, one can explicitly define the RKHS associated with $K$ as

$$\mathcal{H} = \left\{ f = \sum_{\ell=0}^{\infty} \alpha_\ell \phi_\ell : \sum_{\ell=0}^{\infty} \frac{|\alpha_\ell|^2}{\mu_\ell} < \infty \right\}, \tag{59}$$

with the inner-product

$$\left\langle \sum_{\ell=0}^{\infty} \alpha_\ell \phi_\ell, \sum_{\ell=0}^{\infty} \beta_\ell \phi_\ell \right\rangle_{\mathcal{H}} = \sum_{\ell=0}^{\infty} \frac{\alpha_\ell \beta_\ell}{\mu_\ell}, \tag{60}$$

where the sum is considered convergent whenever it converges absolutely. The feature map is therefore given as $\Phi_x = K(x,.) = \sum_{\ell=0}^{\infty} \mu_\ell \phi_\ell(x) \phi_\ell(.)$ for any $x \in \mathcal{M}$. The covariance operator $\Sigma : \mathcal{H} \to \mathcal{H}$ is also defined as $\Sigma = \mathbb{E}_{x \sim \mu}[\Phi_x \otimes_{\mathcal{H}} \Phi_x]$ where the expectation is with respect to the uniform sample $x \in \mathcal{M}$ (with respect to the normalized volume element $d\mu = \frac{1}{\text{vol}(\mathcal{M})} d\,\text{vol}_g(x)$). It is worth mentioning the identity $\|\mathcal{K}^{1/2}(f)\|_{\mathcal{H}} = \|f\|_{L^2(\mathcal{M})}$. Also note if $f^\star = \sum_{\ell=0}^{\infty} \langle f^\star, \phi_\ell \rangle_{L^2(\mathcal{M})} \phi_\ell$, then the effective ridge regression estimator (Equation 4) is given by the closed-form formula

$$\hat{f}_{\text{eff}} = \sum_{\ell=0}^{\infty} \frac{\mu_\ell}{\mu_\ell + \eta} \langle f^\star, \phi_\ell \rangle_{L^2(\mathcal{M})} \phi_\ell. \tag{61}$$

## A.9 Invariant Kernels

A kernel $K : \mathcal{M} \times \mathcal{M} \to \mathbb{R}$ is called $G-$invariant with respect to a closed subgroup $G$ of $\text{ISO}(\mathcal{M})$, if and only if $K(x,y) = K(\tau(x), \tau'(y))$ for any $\tau, \tau' \in G$. Equivalently, one has $K(x,y) = K([x],[y])$ for any $x, y \in \mathcal{M}$. In the previous section, it is observed that $K$ can be written as $K(x,y) = \sum_{\ell=0}^{\infty} \mu_\ell \phi_\ell(x) \phi_\ell(y)$, under a few conditions. Since $K$ is $G-$invariant, a new basis of eigenfunctions exists that allows a more compact representation of the kernel.

**Proposition A.11.** *For any closed subgroup $G$ of $\text{ISO}(\mathcal{M})$, consider a symmetric positive-definite $G-$invariant kernel $K : \mathcal{M} \times \mathcal{M} \to \mathbb{R}$, and assume that $K \in C^2(\mathcal{M} \times \mathcal{M})$ satisfies the differential equation $\Delta_{g,x}(K(x,y)) = \Delta_{g,y}(K(x,y))$. Then, $K$ can be diagonalized in the basis of eigenfunctions of the Laplace-Beltrami operator:*

$$K(x,y) = \sum_{\lambda \in \text{Spec}(\mathcal{M})} \sum_{\ell=1}^{\dim(V_{\lambda,G})} \mu_{\lambda,\ell} \phi_{\lambda,\ell}(x) \phi_{\lambda,\ell}(y), \tag{62}$$

*where the functions $\phi_{\lambda,\ell}$, for any $\lambda \in \text{Spec}(\mathcal{M})$, and any $\ell = 1, \ldots, \dim(V_\lambda)$, form a basis for $L^2(\mathcal{M})$ such that $\Delta_g(\phi_{\lambda,\ell}) + \lambda(\phi_{\lambda,\ell}) = 0$ for each $\ell, \lambda$. Moreover, the functions $\phi_{\lambda,\ell}$, for any $\lambda \in \text{Spec}(\mathcal{M})$, and any $\ell = 1, \ldots, \dim(V_{\lambda,G})$, form an orthonormal basis for $L^2_{\text{inv}}(\mathcal{M}, G)$.*

Therefore, the RKHS of a $G-$invariant kernel $K$ can be defined as

$$\mathcal{H} = \left\{ f = \sum_{\lambda \in \text{Spec}(\mathcal{M})} \sum_{\ell=1}^{\dim(V_{\lambda,G})} \alpha_{\lambda,\ell} \phi_{\lambda,\ell} : \sum_{\lambda \in \text{Spec}(\mathcal{M})} \sum_{\ell=1}^{\dim(V_{\lambda,G})} \frac{|\alpha_{\lambda,\ell}|^2}{\mu_{\lambda,\ell}} < \infty \right\}, \tag{63}$$

with the inner-product

$$\left\langle \sum_{\lambda \in \text{Spec}(\mathcal{M})} \sum_{\ell=1}^{\dim(V_{\lambda,G})} \alpha_{\lambda,\ell} \phi_{\lambda,\ell}, \sum_{\lambda \, \text{Spec}(\mathcal{M})} \sum_{\ell=1}^{\dim(V_{\lambda,G})} \beta_{\lambda,\ell} \phi_{\lambda,\ell} \right\rangle_{\mathcal{H}} = \sum_{\lambda \in \text{Spec}(\mathcal{M})} \sum_{\ell=1}^{\dim(V_{\lambda,G})} \frac{\alpha_{\lambda,\ell} \beta_{\lambda,\ell}}{\mu_{\lambda,\ell}}. \tag{64}$$

Whenever $G$ is the trivial group, the above identities reduce to what is proposed for general (not necessarily invariant) kernels on manifolds in the previous section. Once again, the assumption $K \in C^2(\mathcal{M} \times \mathcal{M})$ can be weakened to just the continuity of $K$.

## A.10 Sobolev Spaces of Functions on Manifolds

For any integer $s \geq 0$, the Sobolev space $\mathcal{H}^s(\mathcal{M})$ is the Hilbert space of measurable functions on $\mathcal{M}$ with square-integrable partial derivatives[5] up to order $s$. More generally, $\mathcal{H}^{s,q}(\mathcal{M})$ denotes the Banach space of measurable functions with $L^p$ bounded partial derivatives up to order $s$. As

---

[5]The partial derivatives on manifolds are defined locally in each coordinate.

observed in [24], one can define the Sobolev space $\mathcal{H}^s(\mathcal{M}) \subset L^2(\mathcal{M})$ using the eigenfunctions of the Laplace-Beltrami operator as

$$\mathcal{H}^s(\mathcal{M}) := \left\{ f = \sum_{\ell=0}^{\infty} \alpha_\ell \phi_\ell : \|f\|_{\mathcal{H}^s(\mathcal{M})}^2 = \sum_{\ell=0}^{\infty} \max(1, \lambda_\ell^s)|\alpha_\ell|^2 < \infty \right\}. \tag{65}$$

The inner-product on $\mathcal{H}^s(\mathcal{M})$ is defined as

$$\left\langle \sum_{\ell=1}^{\infty} \alpha_\ell \phi_\ell, \sum_{\ell=1}^{\infty} \alpha_\ell \phi_\ell \right\rangle_{\mathcal{H}^s(\mathcal{M})} = \sum_{\ell=1}^{\infty} \max(1, \lambda_\ell^s)\alpha_\ell \beta_\ell. \tag{66}$$

This makes $\mathcal{H}^s(\mathcal{M})$ an RKHS with the Sobolev kernel defined as

$$K_{\mathcal{H}^s(\mathcal{M})}(x,y) = \sum_{\lambda \in \mathrm{Spec}(\mathcal{M})} \sum_{\ell=1}^{\dim(V_\lambda)} \min(1, \lambda_\ell^{-s})\phi_{\lambda,\ell}(x)\phi_{\lambda,\ell}(y), \tag{67}$$

For $G-$invariant functions, as before, the above sum must be truncated to $\dim(V_{\lambda,G})$ instead of $\dim(V_\lambda)$. Therefore, $\mathcal{H}_{\mathrm{inv}}^s(\mathcal{M}) = \mathcal{H}^s(\mathcal{M}) \cap L_{\mathrm{inv}}^2(\mathcal{M}, G)$ is well-defined.

We note that $\mathcal{H}^s(\mathcal{M})$ includes only continuous functions when $s > d/2$. Moreover, it contains only continuously differentiable functions up to order $k$ when $s > d/2 + k$; see the Sobolev inequality:

**Proposition A.12** ([3], Sobolev inequality). *Let $\frac{1}{2} - \frac{s}{d} = \frac{1}{q} - \frac{\ell}{d}$ with $s \geq \ell \geq 0$ and $q > 2$, where $d$ is the dimension of the smooth compact closed manifold $\mathcal{M}$. Then,*

$$\|f\|_{\mathcal{H}^{\ell,q}(\mathcal{M})} \leq C\|f\|_{\mathcal{H}^s(\mathcal{M})}. \tag{68}$$

*The constant $C$ may depend only on the manifold and the parameters but is independent of the function $f \in L^2(\mathcal{M})$.*

# B  Proof of Theorem 4.4

We first prove Theorem 4.4 for the cases that the group action on the manifold is *free* (Proposition B.1), and then we extend it to the general case. We use the preset notation/definitions introduced in Appendix A (specifically, Proposition A.8) in this section.

**Proposition B.1.** *Let $(\mathcal{M}, g)$ be a smooth connected compact boundaryless Riemannian manifold of dimension $\dim(\mathcal{M})$. Let $G$ be a Lie subgroup of $\mathrm{ISO}(\mathcal{M})$ of dimension $\dim(G)$, and assume that $G$ acts freely on $\mathcal{M}$ (i.e., having no non-trivial fixed point), and let $d := \dim(\mathcal{M}) - \dim(G)$ denote the effective dimension of the quotient space. Then,*

$$N_x(\lambda; G) := \sum_{\lambda' \leq \lambda} \sum_{\ell=1}^{\dim(V_{\lambda',G})} |\phi_{\lambda',\ell}(x)|^2 = \frac{\omega_d}{(2\pi)^d} \mathrm{vol}(\mathcal{M}/G)\lambda^{d/2} + \mathcal{O}(\lambda^{\frac{d-1}{2}}), \tag{69}$$

*as $\lambda \to \infty$, where $\omega_d = \frac{\pi^{d/2}}{\Gamma(\frac{d}{2}+1)}$ is the volume of the unit $d-$ball in the Euclidean space $\mathbb{R}^d$.*

*Remark* B.2. Note that the above proposition provides a much stronger result than Theorem 4.4; it is *local*. Observe that $N(\lambda; G) = \int_{\mathcal{M}} N_x(\lambda; G)d\,\mathrm{vol}_g(x)$, and thus integrating the left-hand side of the above equation proves Theorem 4.4 for the special case of free actions. We will later prove the same local result (Equation (69)) for the general smooth compact Lie group actions on a manifold, presuming the assumptions in Theorem 4.4.

*Proof of Proposition B.1.* By the quotient manifold theorem (Theorem A.2) and Corollary A.3, the orbit space $\mathcal{M}/G$ is a connected closed (i.e, compact boundaryless) manifold of dimension $d = \dim(\mathcal{M}) - \dim(G)$. Let $\Delta_g$ and $\Delta_{\tilde{g}}$ denote the Laplace-Beltrami operators on $\mathcal{M}$ and $\mathcal{M}/G$, respectively, where $\tilde{g}$ is the induced Riemannian metric on $\mathcal{M}/G$ from $g$. Consider two arbitrary smooth functions $\phi : \mathcal{M} \to \mathbb{R}$ and $\tilde{\phi} : \mathcal{M}/G \to \mathbb{R}$ such that $\phi(x) = \tilde{\phi}([x])$. Note that $\phi$ is smooth on $\mathcal{M}$, if and only if $\tilde{\phi}$ is smooth on $\mathcal{M}/G$, and also, $\phi$ is invariant by definition. Fix an arbitrary $\lambda$. We claim that

$$\Delta_{\tilde{g}}\tilde{\phi} + \lambda\tilde{\phi} = 0 \iff \Delta_g\phi + \lambda\phi = 0 \tag{70}$$

Given the above identity, the desired result follows immediately by an application of the local Weyl's law (Proposition A.1) on the manifold $\mathcal{M}/G$ of dimension $d = \dim(\mathcal{M}) - \dim(G)$.

To prove the claim, we only need to show that $\Delta_{\tilde{g}}\tilde{\phi} = \Delta_g \phi$. First assume that $G$ is a finite group (i.e., $\dim(G) = 0$). Note that in local coordinates $(x^1, x^2, \ldots, x^{\dim(\mathcal{M})})$ we have

$$\Delta_g \phi = \frac{1}{\sqrt{|\det g|}} \partial_i \big( \sqrt{|\det g|} g^{ij} \partial_j \phi \big). \tag{71}$$

However, the projection map $\pi : \mathcal{M} \to \mathcal{M}/G$ is a Riemannian submersion, with differential $d\pi_x : T_x\mathcal{M} \to T_{[x]}(\mathcal{M}/G)$ being an invertible linear map from a $\dim(\mathcal{M})-$dimensional vector space to another $\dim(\mathcal{M})-$dimensional vector space. This shows that the local coordinates $(x^1, x^2, \ldots, x^{\dim(\mathcal{M})})$ are also simultaneously some local coordinates for $\mathcal{M}/G$, and since $\tilde{g}$ is induced by the metric $g$, the result holds by the above identity for the Laplace-Beltrami operator.

Now assume $\dim(G) \geq 1$. In this case, for the projection map $\pi : \mathcal{M} \to \mathcal{M}/G$, the differential map $d\pi_x : T_x\mathcal{M} \to T_{[x]}(\mathcal{M}/G)$ is a surjective linear map from a $\dim(\mathcal{M})-$dimensional vector space to a $(\dim(\mathcal{M}) - \dim(G))-$dimensional vector space. Indeed, $T_x\mathcal{M} = \ker(d\pi_x) \oplus \ker^\perp(d\pi_x)$, with respect to the inner product defined by $g$. This means that with an appropriate choice of local coordinates such as $(x^1, x^2, \ldots, x^{\dim(\mathcal{M})})$ around a point $x \in \mathcal{M}$, satisfying

$$g_{ij} = g\Big(\frac{\partial}{\partial_i}, \frac{\partial}{\partial_j}\Big) = \mathbb{1}\{i = j\}, \tag{72}$$

for any $i, j \in \{1, 2, \ldots, \dim(\mathcal{M})\}$, we have $\frac{\partial}{\partial_i} \in \ker^\perp(d\pi_x)$ for any $i = 1, 2, \ldots, \dim(\mathcal{M}) - \dim(G)$, and $\frac{\partial}{\partial_i} \in \ker(d\pi_x)$ for any $i > \dim(\mathcal{M}) - \dim(G)$. In particular, the restriction of the local coordinates to the first $\dim(\mathcal{M}) - \dim(G)$ elements is assumed to be some local coordinates for $\mathcal{M}/G$. This is always possible for an appropriate choice of local coordinates.

In these specific local coordinates, by definition,

$$\Delta_g \phi = \sum_{i=1}^{\dim(\mathcal{M})} \partial_i^2 \phi \tag{73}$$

$$\Delta_{\tilde{g}}\tilde{\phi} = \sum_{i=1}^{\dim(\mathcal{M})-\dim(G)} \partial_i^2 \tilde{\phi}. \tag{74}$$

Note that $\partial_i^2 \tilde{\phi} = \partial_i^2 \phi$ for $i = 1, 2, \ldots, \dim(\mathcal{M}) - \dim(G)$. Thus, the proof is complete if we show $\partial_i \phi \equiv 0$, for all $i > \dim(\mathcal{M}) - \dim(G)$, for a neighborhood around $x$ in the local coordinates $(x^{\dim(\mathcal{M})-\dim(G)+1}, \ldots, x^{\dim(\mathcal{M})})$, while the other coordinates are kept the same as $x$. But note that for any $x'$ sufficiently close to $x$ with the same coordinates $(x^1, x^2, \ldots, x^{\dim(\mathcal{M})-\dim(G)})$, one has $[x'] = [x]$, by definition. This means that $\phi(x) = \phi(x')$ and this completes the proof. $\qquad\square$

To extend Proposition B.1 to a general smooth compact Lie group action $G$, we need to use the principal orbit theorem (Theorem A.4) and its consequences (see Appendix A.6). Again, we prove that the generalized local result (Equation (69)) holds, presuming the assumptions in Theorem 4.4.

*Proof of Theorem 4.4.* According to Corollary A.5, one has the following decomposition of the quotient space: $\mathcal{M}/G = \bigsqcup_{[H] \preceq G} \mathcal{M}_{[H]}/G$. In other words, the quotient space is the disjoint union of finitely many manifolds, and among them, $\mathcal{M}_0/G$ is open and dense in $\mathcal{M}/G$. As a first step towards the proof, we show that $\Delta_{\tilde{g}}\tilde{\phi} = \Delta_g \phi$ for any two smooth functions $\phi : \mathcal{M}_0 \to \mathbb{R}$ and $\tilde{\phi} : \mathcal{M}_0/G \to \mathbb{R}$ such that $\phi(x) = \tilde{\phi}([x])$, for any $x \in \mathcal{M}_0$ and $[x] \in \mathcal{M}_0/G$. However, the proof of this claim is exactly the same as what is presented in the proof of Proposition B.1; thus, we skip it.

Recall that the effective dimension of the quotient space is defined as $d := \dim(\mathcal{M}_{[H_0]}/G) = \dim(\mathcal{M}_0/G) = \dim(\mathcal{M}) - \dim(G) + \dim(H_0)$, where $[H_0]$ is the unique maximal isotropy type (corresponding to $\mathcal{M}_0$). We claim that there exists a connected compact manifold (possibly with boundary) $\widetilde{\mathcal{M}} \subseteq \mathcal{M}/G$ such that (1) it includes the principal part, i.e., $\widetilde{\mathcal{M}} \supseteq \mathcal{M}_0/G$, and (2) the projected invariant functions on $\widetilde{\mathcal{M}}$ satisfy the Neumann boundary condition on its boundary $\partial(\widetilde{\mathcal{M}})$.

More precisely, the second condition means that for any two smooth functions $\phi : \mathcal{M}_0 \to \mathbb{R}$ and $\tilde{\phi} : \mathcal{M}_0/G \to \mathbb{R}$ such that $\phi(x) = \tilde{\phi}([x])$, for any $x \in \mathcal{M}_0$ and $[x] \in \mathcal{M}_0/G$, the function $\tilde{\phi}$ satisfies the Neumann boundary condition on $\partial(\widetilde{\mathcal{M}})$. Given this claim, by the local Weyl's law (Proposition A.1), the proof is complete.

We only need to specify the manifold $\widetilde{\mathcal{M}}$ and prove that each projected invariant function on it satisfies the Neumann boundary condition. Indeed, the construction of the manifold $\widetilde{\mathcal{M}}$ follows from the finite decomposition of the quotient space $\mathcal{M}/G = \bigsqcup_{[H] \preceq G} \mathcal{M}_{[H]}/G$. Moreover, we can assume that the boundary of $\widetilde{\mathcal{M}}$ is piecewise smooth. Let $[x] \in \partial(\widetilde{\mathcal{M}})$ be a boundary point (in the interior of a smooth piece of the boundary). We claim that

$$\phi \text{ is } G-\text{invariant on } \mathcal{M} \implies \langle \nabla_{\tilde{g}} \tilde{\phi}([x]), \hat{n}_{[x]} \rangle_{\tilde{g}} = 0, \tag{75}$$

for any smooth $\phi : \mathcal{M} \to \mathbb{R}$, where $\phi(x) = \tilde{\phi}([x])$, and this completes the proof. Note that $\hat{n}_{[x]} \in T_{[x]}(\widetilde{\mathcal{M}})$ is the unit outward normal vector of the manifold $\widetilde{\mathcal{M}}$ at $[x]$. To prove the claim, we write $T_{[x]}(\widetilde{\mathcal{M}}) = \text{span}(\hat{n}_{[x]}) \oplus H_{[x]}$ for an orthogonal vector space $H_{[x]}$. But $H_{[x]} \simeq T_{[x]}\partial(\widetilde{\mathcal{M}})$. Also, in a neighborhood $\mathcal{N}$ around $[x]$ in $\partial(\widetilde{\mathcal{M}})$, for each $[y] \in \mathcal{N}$, we have the smooth identity $(\rho_{[y]}^{-1} \circ \tau_{[x]} \circ \rho_{[y]})(y) = y$ for some $\rho_{[y]}, \tau_{[x]} \in G$, such that $(\rho_{[y]}^{-1} \circ \tau_{[x]} \circ \rho_{[y]})$ does not belong to isotropy groups of $\mathcal{M}_0/G$ near $[x]$. Without loss of generality, we assume that $\rho_{[x]} = \text{id}_G$.

Now consider a geodesic on $\mathcal{M}$ starting from $x \in \mathcal{M}$ with unit velocity such as $\gamma(t)$ with $\gamma(0) = x$ and $d\pi_{[x]}(\gamma'(0)) = \hat{n}_{[x]}$. Note that $[\gamma(t)] \notin \partial(\widetilde{M})$ for small enough $t \in (0, \epsilon)$, and thus it belongs to $\mathcal{M}_0/G$. But it is simultaneously "on the other side" of the particular fundamental domain of $\mathcal{M}_0/G$ around $[x]$, meaning that $[\tau_{[x]}(\gamma(t))]$ is necessarily a curve starting from $[x]$ towards the inside of the fundamental domain. In particular, since $\tau_{[x]}$ is an isometry (and thus a local isometry), we necessarily have $(\tau_{[x]} \circ \gamma)'(0) = -\hat{n}_{[x]}$ (note that in this step we clearly use the explanations in the previous paragraph). Now by considering the function $\phi \circ \gamma = \phi \circ \tau_{[x]} \circ \gamma$ on the interval $t \in (0, \epsilon)$, we get

$$\langle \nabla_{\tilde{g}} \tilde{\phi}([x]), \hat{n}_{[x]} \rangle_{\tilde{g}} = \langle \nabla_{\tilde{g}} \tilde{\phi}([x]), -\hat{n}_{[x]} \rangle_{\tilde{g}} \implies \langle \nabla_{\tilde{g}} \tilde{\phi}([x]), \hat{n}_{[x]} \rangle_{\tilde{g}} = 0, \tag{76}$$

and this completes the proof. $\qquad \square$

## C  Proof of Theorem 4.1

In this section, we use Theorem 4.4 to prove Theorem 4.1. Let us first state a standard bound in the literature holding for any RKHS.

**Proposition C.1** ([4], Proposition 7.3). *Consider the KRR problem in an RKHS setting, and let $f^\star_{\text{proj}}$ denote the orthogonal projection of $f^\star$ on the Hilbert space $\mathcal{H}$. Assume that $K(x,x) \leq R^2$ for any $x \in \mathcal{M}$, $\eta \leq R^2$, and $n \geq \frac{5R^2}{\eta}(1 + \log(\frac{R^2}{\eta}))$. Then,*

$$\mathbb{E}[\mathcal{R}(\hat{f}) - \mathcal{R}(f^\star_{\text{proj}})] \leq 16 \frac{\sigma^2}{n} \text{tr}[(\Sigma + \eta I)^{-1}\Sigma] \tag{77}$$

$$+ 16 \inf_{f \in \mathcal{H}} \{ \|f - f^\star_{\text{proj}}\|^2_{L^2(\mathcal{M})} + \eta \|f\|^2_{\mathcal{H}} \} + \frac{24}{n^2} \|f^\star\|^2_{L^\infty(\mathcal{M})}, \tag{78}$$

*where the expectation is over the randomness of the dataset $\mathcal{S}$, and $\Sigma = \mathbb{E}_{x \sim \mu}[\Phi_x \otimes_{\mathcal{H}} \Phi_x]$ is the covariance operator with the feature map $\Phi_x = \sum_{\ell=0}^{\infty} \mu_\ell \phi_\ell(x)\phi_\ell$ for any $x \in \mathcal{M}$.*

Note that $f^\star_{\text{proj}} = f^\star$ if the closure of $\mathcal{H}$ with respect to the $L^2(\mathcal{M})-$norm is $L^2_{\text{inv}}(\mathcal{M}, G)$. In the Laplace-Beltrami basis, $\Sigma$ is diagonal with the diagonal elements $(\Sigma)_{\ell,\ell} = \mu_\ell$ for each $\ell$. Note that the first and second terms in the above upper bound are known as the variance and the bias terms, respectively. Also, while the bound holds in expectation for a random dataset $\mathcal{S}$, assuming $\epsilon_i$'s are sub-Gaussian, one can extend the result to a high-probability bound using standard concentration inequalities. However, for the brevity/clarity of the paper, we restrict our attention to the expectation of the population risk.

We need to have an explicit upper bound for $R$ to use the above proposition. Although the problem is essentially homogenous with respect to $R$, for the sake of completeness, we explicitly compute a uniform upper bound on the diagonal values of the kernel in terms of the problem's parameters. The goal is to first check that the two conditions $R < \infty$ and $n \geq \frac{5R^2}{\eta}(1 + \log(\frac{R^2}{\eta}))$ are satisfied. The latter condition is indeed satisfied when $\eta \geq \frac{5R^2 \log(n)}{n}$. Note that if $\mu_{\lambda,\ell} \neq 0$ for any $\lambda, \ell$ with $\ell = 1, 2, \ldots, \dim(V_{\lambda,G})$, then any $G-$invariant function $f^\star \in \mathcal{F} \subseteq L^2_{\text{inv}}(\mathcal{M}, G)$ is in the closure of $\mathcal{H}$. Indeed, in that case the closure of $\mathcal{H}$ with respect to the $L^2(\mathcal{M})-$norm includes $L^2_{\text{inv}}(\mathcal{M}, G)$.

## C.1  Bounding $K(x, x)$

We start with the definition of $R$; for any $x \in \mathcal{M}$, we have

$$K(x, x) = \langle \Phi_x, \Phi_x \rangle_{\mathcal{H}} = \sum_{\lambda \in \text{Spec}(\mathcal{M})} \sum_{\ell=1}^{\dim(V_{\lambda,G})} \mu_{\lambda,\ell} |\phi_{\lambda,\ell}(x)|^2. \tag{79}$$

By the local version of Theorem 4.4, we know that $N_x(\lambda; G) = \sum_{\lambda' \leq \lambda} \sum_{\ell=1}^{\dim(V_{\lambda',G})} |\phi_{\lambda',\ell}(x)|^2 \leq \frac{\omega_d}{(2\pi)^d} \text{vol}(\mathcal{M}/G)\lambda^{d/2} + C_{\mathcal{M}/G}\lambda^{\frac{d-1}{2}}$, for an absolute constant $C_{\mathcal{M}/G}$, where $d$ denotes the effective dimension of the quotient space. Therefore, if $\mu_{\lambda,\ell} \leq u(\lambda)$ for a differentiable bounded function $u(\lambda)$, for any $\lambda, \ell$, then

$$K(x, x) \leq \int_{0-}^{\infty} u(\lambda) dN_x(\lambda; G) \tag{80}$$

$$\stackrel{(a)}{=} \lim_{\lambda \to \infty} u(\lambda) N_x(\lambda; G) - u(0^-) N_x(0^-; G) - \int_{0-}^{\infty} N_x(\lambda; G) u'(\lambda) d\lambda \tag{81}$$

$$\stackrel{(b)}{\leq} \frac{-\omega_d}{(2\pi)^d} \text{vol}(\mathcal{M}/G) \int_{0-}^{\infty} \lambda^{d/2} u'(\lambda) d\lambda + C_{\mathcal{M}/G} \int_{0-}^{\infty} \lambda^{(d-1)/2} u'(\lambda) d\lambda \tag{82}$$

$$\stackrel{(c)}{=} \frac{\omega_d}{(2\pi)^d} \text{vol}(\mathcal{M}/G) \frac{d}{2} \int_{0-}^{\infty} \lambda^{d/2-1} u(\lambda) d\lambda + C_{\mathcal{M}/G} \int_{0-}^{\infty} \lambda^{(d-1)/2-1} u(\lambda) d\lambda \tag{83}$$

$$= \frac{d}{2} \frac{\omega_d}{(2\pi)^d} \text{vol}(\mathcal{M}/G) \Big( \{\mathcal{M}u\}(d/2) \Big) + C_{\mathcal{M}/G} \Big( \{\mathcal{M}u\}((d-1)/2) \Big), \tag{84}$$

where (a) and (c) follow by integration by parts, and (b) follows from Theorem 4.4. The Mellin transform is defined as $\{\mathcal{M}u\}(s) := \int_0^{\infty} t^{s-1} u(t) dt$. Also, integration with respect to $dN_x(\lambda; G)$ must be understood as a Riemann–Stieltjes integral.

## C.2  Bounding the Bias Term

We have already observed that the function achieving the infimum is

$$\hat{f}_{\text{eff}} = \sum_{\lambda \in \text{Spec}(\mathcal{M})} \sum_{\ell=1}^{\dim(V_{\lambda,G})} \frac{\mu_{\lambda,\ell}}{\mu_{\lambda,\ell} + \eta} \langle f^\star, \phi_{\lambda,\ell} \rangle_{L^2(\mathcal{M})} \phi_{\lambda,\ell}. \tag{85}$$

Note that clearly $\hat{f}_{\text{eff}} \in \mathcal{H}$ for $\eta > 0$, as we can explicitly compute $\|\hat{f}_{\text{eff}}\|_{\mathcal{H}} < \infty$. Also,

$$f^\star_{\text{proj}} = \sum_{\lambda \in \text{Spec}(\mathcal{M})} \sum_{\ell=1}^{\dim(V_{\lambda,G})} \mathbb{1}\{\mu_{\lambda,\ell} \neq 0\} \langle f^\star, \phi_{\lambda,\ell} \rangle_{L^2(\mathcal{M})} \phi_{\lambda,\ell}. \tag{86}$$

Thus,

$$16 \inf_{f \in \mathcal{H}} \left\{ \|f - f^\star_{\text{proj}}\|^2_{L^2(\mathcal{M})} + \eta \|f\|^2_{\mathcal{H}} \right\} = 16 \|\hat{f}_{\text{eff}} - f^\star_{\text{proj}}\|^2_{L^2(\mathcal{M})} + 16\eta \|\hat{f}_{\text{eff}}\|^2_{\mathcal{H}} \tag{87}$$

$$= 16 \sum_{\lambda \in \text{Spec}(\mathcal{M})} \sum_{\ell=1}^{\dim(V_{\lambda,G})} \left( \frac{\mu_{\lambda,\ell}}{\mu_{\lambda,\ell} + \eta} - 1 \right)^2 \langle f^\star_{\text{proj}}, \phi_{\lambda,\ell} \rangle^2_{L^2(\mathcal{M})} \tag{88}$$

$$+ 16\eta \sum_{\lambda \in \mathrm{Spec}(\mathcal{M})} \sum_{\ell=1}^{\dim(V_{\lambda,G})} \frac{1}{\mu_{\lambda,\ell}} \left( \frac{\mu_{\lambda,\ell}}{\mu_{\lambda,\ell} + \eta} \langle f^\star_{\mathrm{proj}}, \phi_{\lambda,\ell} \rangle_{L^2(\mathcal{M})} \right)^2 \quad (89)$$

$$= 16 \sum_{\lambda \in \mathrm{Spec}(\mathcal{M})} \sum_{\ell=1}^{\dim(V_{\lambda,G})} \left( \frac{\eta^2 + \eta\mu_{\lambda,\ell}}{(\mu_{\lambda,\ell} + \eta)^2} \right) \langle f^\star_{\mathrm{proj}}, \phi_{\lambda,\ell} \rangle_{L^2(\mathcal{M})}^2 \quad (90)$$

$$= 16\eta \sum_{\lambda \in \mathrm{Spec}(\mathcal{M})} \sum_{\ell=1}^{\dim(V_{\lambda,G})} \left( \frac{1}{\mu_{\lambda,\ell} + \eta} \right) \langle f^\star_{\mathrm{proj}}, \phi_{\lambda,\ell} \rangle_{L^2(\mathcal{M})}^2. \quad (91)$$

### C.3 Bounding the Variance Term

We need to compute the trace of the operator $(\Sigma + \eta I)^{-1}\Sigma$. But this operator is diagonal in the Laplace-Beltrami basis, and thus we get

$$16\frac{\sigma^2}{n} \mathrm{tr}[(\Sigma + \eta I)^{-1}\Sigma] = 16\frac{\sigma^2}{n} \sum_{\lambda \in \mathrm{Spec}(\mathcal{M})} \sum_{\ell=1}^{\dim(V_{\lambda,G})} \frac{\mu_{\lambda,\ell}}{\mu_{\lambda,\ell} + \eta}. \quad (92)$$

### C.4 Bounding the Population Risk

Now we combine the previous steps to get

$$\mathbb{E}[\mathcal{R}(\hat{f}) - \mathcal{R}(f^\star_{\mathrm{proj}})] \leq 16\frac{\sigma^2}{n} \sum_{\lambda \in \mathrm{Spec}(\mathcal{M})} \sum_{\ell=1}^{\dim(V_{\lambda,G})} \frac{\mu_{\lambda,\ell}}{\mu_{\lambda,\ell} + \eta} \quad (93)$$

$$+ 16\eta \sum_{\lambda \in \mathrm{Spec}(\mathcal{M})} \sum_{\ell=1}^{\dim(V_{\lambda,G})} \left( \frac{1}{\mu_{\lambda,\ell} + \eta} \right) \langle f^\star_{\mathrm{proj}}, \phi_{\lambda,\ell} \rangle_{L^2(\mathcal{M})}^2 \quad (94)$$

$$+ \frac{24}{n^2} \|f^\star\|_{L^\infty(\mathcal{M})}^2, \quad (95)$$

which holds when $R < \infty$ and $\eta \geq \frac{5R^2 \log(n)}{n}$.

We are now ready to bound the convergence rate of the population risk of KRR for invariant Sobolev space $\mathcal{H}^s_{\mathrm{inv}}(\mathcal{M})$ (See Section A.10 for the definition). In this case, $\mu_{\lambda,\ell} = u(\lambda) = \min(1, \lambda^{-s})$ for each $\lambda, \ell$. Therefore,

$$\{\mathcal{M}u\}(d/2) = \int_0^\infty \min(1, \lambda^{-s}) t^{d/2-1} dt \leq 1 + \frac{1}{s - d/2}. \quad (96)$$

Similarly, $\{\mathcal{M}u\}((d-1)/2) \leq 1 + \frac{1}{s-(d-1)/2}$. Thus, using the analysis in Section C.1, we get

$$K_{\mathcal{H}^s(\mathcal{M})}(x,x) \leq R^2 := \frac{\omega_d}{(2\pi)^d} \mathrm{vol}(\mathcal{M}/G) \left( d/2 + \frac{d/2}{s - d/2} \right) \quad (97)$$

$$+ C_{\mathcal{M}/G} \left( 1 + \frac{1}{s - (d-1)/2} \right). \quad (98)$$

In particular, $R < \infty$ if $s > d/2$. We now compute the bias and the variance terms as follows. Let us start with the variance term:

$$16\frac{\sigma^2}{n} \sum_{\lambda \in \mathrm{Spec}(\mathcal{M})} \sum_{\ell=1}^{\dim(V_{\lambda,G})} \frac{\mu_{\lambda,\ell}}{\mu_{\lambda,\ell} + \eta} = 16\frac{\sigma^2}{n} \int_{0-}^\infty \frac{\min(1, \lambda^{-s})}{\min(1, \lambda^{-s}) + \eta} dN(\lambda; G) \quad (99)$$

$$\leq 16\frac{\sigma^2}{n} N(1; G) + 16\frac{\sigma^2}{n} \int_1^\infty \frac{\lambda^{-s}}{\lambda^{-s} + \eta} dN(\lambda; G) \quad (100)$$

$$= 16\frac{\sigma^2}{n} N(1; G) + 16\frac{\sigma^2}{n} \int_1^\infty \frac{1}{1 + \eta\lambda^s} dN(\lambda; G) \quad (101)$$

$$\overset{(a)}{=} 16\frac{\sigma^2}{n}N(1;G) + 16\frac{\sigma^2}{n}\lim_{\lambda\to\infty}\left(\frac{N(\lambda;G)}{1+\eta\lambda^s}\right) - 16\frac{\sigma^2}{n}\frac{N(1;G)}{1+\eta} \tag{102}$$

$$+ 16\frac{\sigma^2}{n}\int_1^\infty N(\lambda;G)\frac{s\eta\lambda^{s-1}}{(1+\eta\lambda^s)^2}d\lambda \tag{103}$$

$$\overset{(b)}{=} 16\frac{\sigma^2}{n}\frac{\eta}{1+\eta}N(1;G) + 16\frac{\sigma^2}{n}\int_1^\infty N(\lambda;G)\frac{s\eta\lambda^{s-1}}{(1+\eta\lambda^s)^2}d\lambda \tag{104}$$

$$\overset{(c)}{=} 16\frac{\sigma^2}{n}\frac{\eta}{1+\eta}N(1;G) + 16\frac{\sigma^2}{n}\frac{\omega_d}{(2\pi)^d}\mathrm{vol}(\mathcal{M}/G)s\eta\int_1^\infty \lambda^{d/2}\frac{\lambda^{s-1}}{(1+\eta\lambda^s)^2}d\lambda \tag{105}$$

$$+ 16\frac{\sigma^2}{n}C_{\mathcal{M}/G}s\eta\int_1^\infty \lambda^{(d-1)/2}\frac{\lambda^{s-1}}{(1+\eta\lambda^s)^2}d\lambda \tag{106}$$

$$\leq 16\frac{\sigma^2}{n}\eta N(1;G) + 16\frac{\sigma^2}{n}\frac{\omega_d}{(2\pi)^d}\mathrm{vol}(\mathcal{M}/G)s\eta\int_1^\infty \frac{\lambda^{d/2+s-1}}{1+\eta^2\lambda^{2s}}d\lambda \tag{107}$$

$$+ 16\frac{\sigma^2}{n}C_{\mathcal{M}/G}s\eta\int_1^\infty \frac{\lambda^{(d-1)/2+s-1}}{1+\eta^2\lambda^{2s}}d\lambda, \tag{108}$$

where (a) follows by integration by parts, (b) follows by $\lim_{\lambda\to\infty}\left(\frac{N(\lambda;G)}{1+\eta\lambda^s}\right) = 0$ since $s > d/2$, and (c) follows from Theorem 4.4. Note that integration with respect to $dN(\lambda;G)$ must be understood as a Riemann–Stieltjes integral. By a change of variable in the integrals as $t = \lambda\eta^{1/s}$, we have

$$\int_1^\infty \frac{\lambda^{d/2+s-1}}{1+\eta^2\lambda^{2s}}d\lambda = \eta^{\frac{-1}{s}(s+d/2-1)}\eta^{-1/s}\int_1^\infty \frac{t^{d/2+s-1}}{1+t^{2s}}dt \leq \frac{\eta^{\frac{-1}{s}(s+d/2)}}{2s-d}. \tag{109}$$

We can similarly evaluate the other integral and conclude

$$16\frac{\sigma^2}{n}\sum_{\lambda\in\mathrm{Spec}(\mathcal{M})}\sum_{\ell=1}^{\dim(V_{\lambda,G})}\frac{\mu_{\lambda,\ell}}{\mu_{\lambda,\ell}+\eta} \leq 16\frac{\sigma^2}{n}\eta N(1;G) \tag{110}$$

$$+ 16\frac{s\sigma^2}{(2s-d)n}\frac{\omega_d}{(2\pi)^d}\mathrm{vol}(\mathcal{M}/G)\eta^{1-\frac{1}{s}(s+d/2)} \tag{111}$$

$$+ 16\frac{s\sigma^2}{(2s-d+1)n}C_{\mathcal{M}/G}\eta^{1-\frac{1}{s}(s+(d-1)/2)}. \tag{112}$$

One can also use the bound $N(1;G) \leq \frac{\omega_d}{(2\pi)^d}\mathrm{vol}(\mathcal{M}/G) + C_{\mathcal{M}/G}$ to simplify the upper bound.

Note that $f^\star_{\mathrm{proj}} = f^\star$ since the closure of $\mathcal{H}^s_{\mathrm{inv}}(\mathcal{M})$ with respect to the $L^2(\mathcal{M})-$norm is the whole space $L^2_{\mathrm{inv}}(\mathcal{M},G)$. Let us now analyze the bias term by noting that $\mu_{\lambda,\ell}+\eta \geq \mu_{\lambda,\ell}^\theta\eta^{1-\theta}$ for any $\theta\in(0,1]$, and thus

$$16\eta\sum_{\lambda\in\mathrm{Spec}(\mathcal{M})}\sum_{\ell=1}^{\dim(V_{\lambda,G})}\left(\frac{1}{\mu_{\lambda,\ell}+\eta}\right)\langle f^\star, \phi_{\lambda,\ell}\rangle_{L^2(\mathcal{M})}^2 \tag{113}$$

$$\leq 16\eta^\theta\sum_{\lambda\in\mathrm{Spec}(\mathcal{M})}\sum_{\ell=1}^{\dim(V_{\lambda,G})}\mu_{\lambda,\ell}^{-\theta}\langle f^\star, \phi_{\lambda,\ell}\rangle_{L^2(\mathcal{M})}^2 \tag{114}$$

$$= 16\eta^\theta\sum_{\lambda\in\mathrm{Spec}(\mathcal{M})}\sum_{\ell=1}^{\dim(V_{\lambda,G})}\max(1,\lambda^{s\theta})\langle f^\star, \phi_{\lambda,\ell}\rangle_{L^2(\mathcal{M})}^2 \tag{115}$$

$$= 16\eta^\theta\|f^\star\|_{\mathcal{H}^{s\theta}_{\mathrm{inv}}(\mathcal{M})}^2, \tag{116}$$

where $\theta\in(0,1]$ is chosen so that $f^\star\in\mathcal{H}^{s\theta}_{\mathrm{inv}}(\mathcal{M})$. Therefore,

$$\mathbb{E}[\mathcal{R}(\hat{f})-\mathcal{R}(f^\star)] \leq 16\frac{\sigma^2}{n}\eta\left(\frac{\omega_d}{(2\pi)^d}\mathrm{vol}(\mathcal{M}/G)+C_{\mathcal{M}/G}\right) \tag{117}$$

$$+ 16 \frac{s\sigma^2}{(2s-d)n} \frac{\omega_d}{(2\pi)^d} \operatorname{vol}(\mathcal{M}/G)\eta^{1-\frac{1}{s}(s+d/2)} \tag{118}$$

$$+ 16 \frac{s\sigma^2}{(2s-d+1)n} C_{\mathcal{M}/G}\eta^{1-\frac{1}{s}(s+(d-1)/2)} \tag{119}$$

$$+ 16\eta^\theta \|f^\star\|^2_{\mathcal{H}^{s\theta}_{\mathrm{inv}}(\mathcal{M})} + \frac{24}{n^2}\|f^\star\|^2_{L^\infty(\mathcal{M})}, \tag{120}$$

The result is only true when $R < \infty$ and $\eta \geq \frac{5R^2 \log(n)}{n}$, where $R$ is defined in Equation (98).

We can now optimize the above bound for $\eta$. First consider the function $p(t) = c_a t^{-a} + c_b t^b$ on $\mathbb{R}_{\geq 0}$ for $a, b, c_a, c_b > 0$. Note that $p(0) = p(\infty) = \infty$, and to find its stationary points, we solve $p'(t) = 0$ and get the only solution $t = (\frac{ac_a}{bc_b})^{1/(a+b)}$ which is thus the global minimum of the function. Thus by minimizing

$$p(\eta) = \Big( 16\frac{s\sigma^2}{(2s-d)n} \frac{\omega_d}{(2\pi)^d} \operatorname{vol}(\mathcal{M}/G)\Big)\eta^{-\frac{d}{2s}} + \Big( 16\|f^\star\|^2_{\mathcal{H}^{s\theta}_{\mathrm{inv}}(\mathcal{M})}\Big)\eta^\theta, \tag{121}$$

we get

$$\eta = \Big( \frac{d\sigma^2}{2\theta\|f^\star\|^2_{\mathcal{H}^{s\theta}_{\mathrm{inv}}(\mathcal{M})}(2s-d)n} \frac{\omega_d}{(2\pi)^d} \operatorname{vol}(\mathcal{M}/G)\Big)^{s/(\theta s+d/2)}. \tag{122}$$

Therefore, the population risk is bounded as follows:

$$\mathbb{E}[\mathcal{R}(\hat{f}) - \mathcal{R}(f^\star)] \tag{123}$$

$$\leq \underbrace{16\Big(\frac{\omega_d}{(2\pi)^d}\operatorname{vol}(\mathcal{M}/G) + C_{\mathcal{M}/G}\Big)\frac{\sigma^2}{n}\Big(\frac{d\sigma^2}{2\theta\|f^\star\|^2_{\mathcal{H}^{s\theta}_{\mathrm{inv}}(\mathcal{M})}(2s-d)n}\frac{\omega_d}{(2\pi)^d}\operatorname{vol}(\mathcal{M}/G)\Big)^{s/(\theta s+d/2)}}_{\mathcal{O}(n^{-1-s/(\theta s+d/2)})} \tag{124}$$

$$+ \underbrace{16\frac{s\sigma^2}{(2s-d+1)n}C_{\mathcal{M}/G}\Big(\frac{d\sigma^2}{2\theta\|f^\star\|^2_{\mathcal{H}^{s\theta}_{\mathrm{inv}}(\mathcal{M})}(2s-d)n}\frac{\omega_d}{(2\pi)^d}\operatorname{vol}(\mathcal{M}/G)\Big)^{-(d-1)/(2\theta s+d)}}_{\mathcal{O}(n^{-(\theta s+1/2)/(\theta s+d/2)})} \tag{125}$$

$$+ \underbrace{32\Big(\frac{d\sigma^2}{2\theta(2s-d)n}\frac{\omega_d}{(2\pi)^d}\operatorname{vol}(\mathcal{M}/G)\Big)^{\theta s/(\theta s+d/2)}\|f^\star\|^{d/(\theta s+d/2)}_{\mathcal{H}^{s\theta}_{\mathrm{inv}}(\mathcal{M})}}_{\mathcal{O}(n^{-\theta s/(\theta s+d/2)})} \tag{126}$$

$$+ \underbrace{\frac{24}{n^2}\|f^\star\|^2_{L^\infty(\mathcal{M})}}_{\mathcal{O}(1/n^2)}, \tag{127}$$

and this completes the proof since $s = \frac{d}{2}(\kappa + 1)$ and the third term dominates the sum.

## D Proofs of Theorem 4.6, Proposition 4.5, and Corollary 4.7

Corollary 4.7 follows from Theorem 4.6 on the dimension of the vector space $\mathcal{H}_G$, just according to standard bounds in the literature on the population risk of KRR for finite-rank kernels (see [65]). Therefore, we focus on the proof of Theorem 4.6 and Proposition 4.5.

*Proof of Proposition 4.5.* This is a classical result; one can use a variational method to prove it. See [12] for the proof for an arbitrary vector space $V$. Here, we prove it for $V = \mathcal{H}_G$. Consider an arbitrary $f = \sum_{\ell=0}^{D-1} \langle f, \phi_\ell \rangle_{L^2(\mathcal{M})}\phi_\ell \in \mathcal{H}_G$. Then, using Equation (27),

$$\mathcal{E}(f, f) = \int_{\mathcal{M}} |\nabla_g f(x)|^2_g d\mathrm{vol}_g(x) \tag{128}$$

$$= \sum_{\ell=0}^{D-1} \lambda_\ell |\langle f, \phi_\ell \rangle_{L^2(\mathcal{M})}|^2 \tag{129}$$

$$\leq \lambda_{D-1} \sum_{\ell=0}^{D-1} \langle f, \phi_\ell \rangle_{L^2(\mathcal{M})}^2 \tag{130}$$

$$= \lambda_{D-1} \|f\|_{L^2(\mathcal{M})}^2, \tag{131}$$

and the bound is achieved by $f = \phi_{D-1}$. $\qquad\square$

Now by Theorem 4.4, the proof of Theorem 4.6 is also complete.

## E  Proof of Theorem 4.3

In this section, we mostly use/follow standard results in the literature of minimax lower bounds which can be found in [65]. Note that the unit ball in the Sobolev space $\mathcal{H}_{\text{inv}}^s(\mathcal{M})$ is isomorphic to the following ellipsoid (see Appendix A.10):

$$\mathcal{E} := \left\{ (\alpha_\ell)_{\ell=0}^\infty : \sum_{\ell=0}^\infty \frac{|\alpha_\ell|^2}{\min(1, \lambda_\ell^{-s})} \leq 1 \right\} \subseteq \ell^2(\mathbb{N}). \tag{132}$$

Note that the eigenvalues are distributed according to the bound proved in Theorem 4.4. Consider $M$ functions/sequences $f_1, f_2, \ldots, f_M \in \mathcal{E}$ such that $\|f_i - f_j\|_{\ell^2(\mathbb{N})} \geq \delta$ for all $i \neq j$, for some $M$ and $\delta$ (to be set later). In other words, let $\{f_1, f_2, \ldots, f_M\}$ denote a $\delta-$packing of the set $\mathcal{E}$ in the $\ell^2(\mathbb{N})-$norm.

Consider a pair of random variables $(Z, J)$ as follows: first $J \in [M] := \{1, 2, \ldots, M\}$ and $x_i \in \mathcal{M}$, $i = 1, 2 \ldots, n$, are chosen uniformly and independently at random, and then, $Z = (f_J(x_i) + \epsilon_i)_{i=1}^n \in \mathbb{R}^n$, where $\epsilon_i \sim \mathcal{N}(0, \sigma^2)$ are independent Gaussian variates. Let $\mathbb{P}_j(.) = \mathbb{P}(.|J = j)$ denote the conditional law of $(Z, J)$, given the observation $J = j$. A straighforward computation shows that $D_{\text{KL}}(\mathbb{P}_i || \mathbb{P}_j) = \frac{n}{2\sigma^2} \|f_i - f_j\|_{\ell^2(\mathbb{N})}^2 \geq \frac{n\delta}{2\sigma^2}$ for all $i, j \in [M]$.

According to Fano's method, one can get the minimax bound

$$\inf_{\hat{f}} \sup_{\substack{f^\star \in \mathcal{H}_{\text{inv}}^s(\mathcal{M}) \\ \|f^\star\|_{\mathcal{H}_{\text{inv}}^s(\mathcal{M})}=1}} \mathbb{E}\Big[ \mathcal{R}(\hat{f}) - \mathcal{R}(f^\star) \Big] \geq \frac{1}{2}\delta^2, \tag{133}$$

if $\log(M) \geq 2I(Z; J) + 2\log(2)$. Using the Yang-Barron method [65], this condition is satisfied if

$$\epsilon^2 \geq \log N_{\text{KL}}(\epsilon) \tag{134}$$

$$\log M \geq 4\epsilon^2 + 2\log(2), \tag{135}$$

where $N_{\text{KL}}(\epsilon)$ denotes the $\epsilon-$covering number of the space of distributions $\mathbb{P}(.|f)$ for some $f \in \mathcal{E}$ (defined similarly as we have $\mathbb{P}(.|J = j) = \mathbb{P}(.|f = f_j)$), in the square-root KL-divergence. However, since $D_{\text{KL}}(\mathbb{P}_f || \mathbb{P}_g) = \frac{n}{2\sigma^2} \|f - g\|_{\ell^2(\mathbb{N})}^2$, this equals to the $\epsilon-$covering of the space $\mathcal{E}$ in the $\ell^2(\mathbb{N})-$norm. In other words, we have

$$N_{\text{KL}}(\epsilon) = N_{\ell^2(\mathbb{N})}\Big( \frac{\epsilon \sigma \sqrt{2}}{\sqrt{n}} \Big). \tag{136}$$

Now note that, for any $M$ such that $\log M \geq \log N_{\ell^2(\mathbb{N})}(\delta)$, there exists a $\delta-$packing of the space $\mathcal{E}$ in the $\ell^2(\mathbb{N})-$norm (see the packing and covering numbers relationship [65]).

In summary, we get the minimax rate of $\frac{1}{2}\delta^2$ if the following inequalities are satisfied:

$$\epsilon^2 \geq \log N_{\ell^2(\mathbb{N})}\Big( \frac{\epsilon \sigma \sqrt{2}}{\sqrt{n}} \Big) \tag{137}$$

$$\log N_{\ell^2(\mathbb{N})}(\delta) \geq 4\epsilon^2 + 2\log(2), \tag{138}$$

for some pair $(\epsilon, \delta)$. Thus, our goal is to obtain tight lower/upper bounds for $\log N_{\ell^2(\mathbb{N})}(.)$.

**Lemma E.1.** *For any positive $\zeta$,*

$$\log N_{\ell^2(\mathbb{N})}(\zeta/2) \geq N(\zeta^{\frac{-2}{s}}; G) \log(2) \tag{139}$$

$$\log N_{\ell^2(\mathbb{N})}(\sqrt{2}\zeta) \leq N(\zeta^{\frac{-2}{s}}; G)(s/d + \log(4)) + \mathcal{O}(\zeta^{-\frac{d-1}{s}}), \tag{140}$$

*where the quantity $N(\zeta^{\frac{-2}{s}}; G)$ is defined in Theorem 4.4.*

First, let us show how the above lemma concludes the proof of Theorem 4.3. According to the lemma, we just need to check the following inequalities:

$$\epsilon^2 \geq (s/d + \log(4))\frac{\omega_d}{(2\pi)^d} \operatorname{vol}(\mathcal{M}/G)\left(\frac{\epsilon\sigma}{\sqrt{n}}\right)^{-d/s} + \mathcal{O}(n^{-\frac{d-1}{2s}}) \tag{141}$$

$$N((2\delta)^{\frac{-2}{s}}; G)\log(2) \geq 4\epsilon^2 + 2\log(2). \tag{142}$$

Without loss of generality, let us discard the big-O error terms in the above analysis. In our final adjustment, we can add a constant multiplicative factor to ensure that the bound is asymptotically valid. To get the largest possible $\delta$, we set

$$\epsilon^2 = (s/d + \log(4))\frac{\omega_d}{(2\pi)^d} \operatorname{vol}(\mathcal{M}/G)\left(\frac{\epsilon\sigma}{\sqrt{n}}\right)^{-d/s}, \tag{143}$$

and thus

$$\epsilon^2 = \left((s/d + \log(4))\frac{\omega_d}{(2\pi)^d} \operatorname{vol}(\mathcal{M}/G)\right)^{s/(s+d/2)} \times \left(\frac{\sigma^2}{n}\right)^{-d/(s+d/2)}. \tag{144}$$

Therefore, the following inequality needs to be satisfied:

$$N((2\delta)^{\frac{-2}{s}}; G)\log(2) \geq \log(4) + 4\left((s/d + \log(4))\frac{\omega_d}{(2\pi)^d} \operatorname{vol}(\mathcal{M}/G)\right)^{s/(s+d/2)}\left(\frac{\sigma^2}{n}\right)^{-d/(s+d/2)}. \tag{145}$$

Using asymptotic analysis, the inequality holds when

$$\frac{\log(2)\omega_d}{(2\pi)^d} \operatorname{vol}(\mathcal{M}/G)(2\delta)^{-d/s} \geq 4\left((s/d + \log(4))\frac{\omega_d}{(2\pi)^d} \operatorname{vol}(\mathcal{M}/G)\right)^{s/(s+d/2)}\left(\frac{\sigma^2}{n}\right)^{-d/(s+d/2)}. \tag{146}$$

Rearranging the terms shows that

$$4\delta^2 \leq \left(\frac{\omega_d}{(2\pi)^d}\frac{\sigma^2 \operatorname{vol}(\mathcal{M}/G)}{n}\right)^{s/(s+d/2)} \times \underbrace{\left(\frac{4}{\log(2)\left(s/d + 2\log(2)\right)^{-s/(s+d/2)}}\right)^{-2s/d}}_{:=8C_\kappa}, \tag{147}$$

where $C_\kappa$ only depends on $\kappa = 2s/d - 1$. Since this gives a minimax lower bound of $\frac{1}{2}\delta^2$, the proof is complete.

The rest of this section is devoted to the proof of Lemma E.1.

*Proof of Lemma E.1.* Define the following truncated ellipsoid:

$$\tilde{\mathcal{E}} := \left\{(\alpha_\ell)_{\ell=0}^\infty \in \mathcal{E} : \forall \ell \geq \Delta + 1 : \alpha_\ell = 0\right\} \subseteq \mathcal{E}, \tag{148}$$

where $\Delta$ is a parameter defined as

$$\Delta := \max\left\{\ell : \lambda_\ell \leq \zeta^{\frac{-2}{s}}\right\} = N(\zeta^{\frac{-2}{s}}; G). \tag{149}$$

Note that $\sum_{\ell=\Delta+1}^\infty |\alpha_\ell|^2 \leq \zeta^2$ for all $(\alpha_\ell)_{\ell=0}^\infty \in \mathcal{E}$.

To construct a $\zeta/2-$covering set, note that according to the definition of the truncated ellipsoid, $\mathcal{B}_\Delta(\zeta) \subseteq \tilde{\mathcal{E}} \subseteq \mathcal{E}$, where $\mathcal{B}_\Delta(\zeta)$ denotes the ball with radius $\zeta$ (in $\ell^2-$norm) is the Euclidean space of dimension $\Delta$. Using standard bounds in the literature [65], we get

$$\log N_{\ell^2(\mathbb{N})}(\zeta/2) \geq \Delta \log(2). \tag{150}$$

To get the upper bound on the $\sqrt{2}\zeta-$covering number, using an argument based on the volume of the ellipsoid $\tilde{\mathcal{E}}$ [65], we conclude

$$\log N_{\ell^2(\mathbb{N})}(\sqrt{2}\zeta) \leq \Delta \log(4/\zeta) + \frac{1}{2}\sum_{\ell=0}^{\Delta}\log(\lambda_\ell^{-s}) \tag{151}$$

$$= \Delta \log(4/\zeta) - \frac{s}{2}\sum_{\ell=0}^{\Delta}\log(\lambda_\ell) \tag{152}$$

$$= \Delta \log(4/\zeta) - \frac{s}{2}\int_1^{\zeta^{\frac{-2}{s}}}\log(t)dN(t;G) \tag{153}$$

$$= \Delta \log(4/\zeta) - \frac{s}{2}\log(\zeta^{\frac{-2}{s}})N(\zeta^{\frac{-2}{s}};G) + \frac{s}{2}\int_1^{\zeta^{\frac{-2}{s}}}N(t;G)\frac{dt}{t} \tag{154}$$

$$= \Delta \log(4) + \frac{s}{2}\int_1^{\zeta^{\frac{-2}{s}}}N(t;G)\frac{dt}{t}. \tag{155}$$

By Theorem 4.4,

$$\int_1^{\zeta^{\frac{-2}{s}}}N(t;G)\frac{dt}{t} \leq \frac{2}{d}N(\zeta^{\frac{-2}{s}};G) + \mathcal{O}(\zeta^{-\frac{d-1}{s}}) \tag{156}$$

$$= \frac{2}{d}\frac{\omega_d}{(2\pi)^d}\operatorname{vol}(\mathcal{M}/G)\zeta^{-d/s} + \mathcal{O}(\zeta^{-\frac{d-1}{s}}). \tag{157}$$

Therefore,

$$\log N_{\ell^2(\mathbb{N})}(\sqrt{2}\zeta) \leq \Delta(s/d + \log(4)) + \mathcal{O}(\zeta^{-\frac{d-1}{s}}). \tag{158}$$

$\square$

