# OpenReview forum: "The Exact Sample Complexity Gain from Invariances for Kernel Regression"
_NeurIPS.cc/2023/Conference — NeurIPS 2023 spotlight_

### Official Review · Reviewer_DGX1 · 2023-07-05

**Soundness:** 3 good
**Presentation:** 3 good
**Contribution:** 4 excellent
**Rating:** 7
**Confidence:** 3

**Summary:**

This paper analyzes the impact on sample complexity of encoding invariances into kernel functions in the context of kernel ridge regression. A kernel function is invariant to the actions of a group if its output does not change when its inputs are acted on by members of the group. The paper shows that for finite groups, the sample efficiency is effectively multiplied by the size of the group. For groups of positive dimension, it shows the sample efficiency is improved by (1) reducing the effective dimension of the input manifold, and (2) reducing the volume of the manifold.


**Strengths:**

- Significance: This paper analyzes how invariance improves sample complexity for kernel ridge regression in a much more general setting than prior work. For example, it allows the input space to be any compact manifold, and the invariance to be represented by any smooth compact Lie group (prior work, for example [5], had assumed the input manifold was a sphere and that the invariance was to a finite group).
- Quality: Although I have not checked the proofs, the quality of the work appears high. The generalization bound in Theorem 3.1 is proved to be minimax optimal in Theorem 3.3, an interesting/important result.
- Clarity: The paper is generally well written and clear. See “Weaknesses” section for places for improvement.
- Originality:  It seems the approach of using differential geometry to analyze these generalization bounds is original, and that the authors developed some tools of independent interest in this area.

Overall, I believe this paper makes an important theoretical contribution to a problem space that has been of significant interest to the machine learning community: namely, how to make models invariant to certain input transformation (either via data augmentation, or by changing model architecture), and understanding how encoding these invariances impacts the models’ generalization performance and sample complexity.


**Weaknesses:**

Perhaps the greatest weakness of this paper is at times seeming too “mathy” (https://arxiv.org/pdf/1807.03341.pdf), without sufficient grounding in concrete examples or empirical results. For example:
- It would have been useful to go over more examples of invariances whose groups have positive dimension, and understand more concretely how these invariances affect the generalization bound.
- It could be useful, for equations (5) and (7), to give examples of how this quantity changes for different finite vs. infinite (positive dimension) groups. One example I had in mind, for $x \in R^2$, was rotational invariance, where the rotations could be in a finite group (e.g., rotate by multiples of $\alpha$ degrees, for any $\alpha$ that evenly divides 360) or an infinite group (e.g., rotate by $\beta$ degrees for any real-value $\beta \in [0, 360)$). How would these situations compare, in terms of equations (5) and (7), as the value of $\alpha$ approaches 0?
- It would have been helpful to understand which of the examples given would not have been possible to analyze with prior results.
- It would have been useful to run experiments (even toy/synthetic experiments if necessary) to demonstrate that the bounds from the paper are actually predictive of model performance.

NIT: $\omega_d$ is defined in Theorem 3.4, but used in prior theorems.

**Questions:**

See the weaknesses section.

**Limitations:**

Perhaps a broader discussion of limitations would be useful.
- What deep learning scenarios would the results from this paper apply to or not apply to?
- What important open problems remain to be solved?
- How empirically predictive are these theoretical results, in different settings? In what cases would these results definitely not be empirically predictive?

---

> ### Author Rebuttal · Authors · 2023-08-10
>
>
>  We thank the respected reviewer for their helpful comments. Here we provide a response to the questions/weaknesses mentioned by the reviewer.
>
>
> - Question: "It would ... generalization bound."
>
>   - Answer: We thank the reviewer for their comment. While we devoted a full section to examples and applications (Section 4 in the main body of the paper), we appreciate the reviewer's suggestion and as there are many examples of learning under invariances (e.g., in physics) with different groups, we will survey even more in our next version to clarify the various kinds of gains available according to our results.
>
>  - Question: "It could ... approaches 0?"
>
>
>      - Answer: We will add the example provided by the reviewer to the paper and add more explanations since it allows easy computation of the gain and makes the paper more readable.  Indeed, since we are going to approximate a one-dimensional group using finite groups, the gain is in the exponent and thus the one-dimensional group is always better (for large sample sizes).
>
>
>   - Question: "It would ... prior results."
>
>       - Answer: Since all the provided examples are not using spheres, they are not possible to be analyzed using the prior work. We will clarify this in the next version of the paper.
>
>
>   - Question: "It would ... model performance."
>
>       - Answer: Thanks for making this suggestion. Although we believe that the results are theoretical in nature, we plan to add supporting experiments to the next version of the paper.
>
>
>   - Question: "NIT ... theorems"
>
>       - Answer: We will clarify it in the next version of the paper.
>
>  - Question: "What deep learning ... to?"
>
>      - Answer: The results apply to the kernel approximations of deep networks (e.g., lazy regimes), but for general deep learning it's still an open problem. We will clarify this in the paper.
>
>  - Question: "What important open problems remain to be solved?"
>
>      - Answer: Perhaps the main open problem is to generalize the results of the paper to sample the complexity of other statistical estimation problems  (e.g., Wasserstein distance, density estimation, etc.) under invariances on manifolds. Also, it's interesting to exploit the possibility of applying the results to NTK regime for invariant models. We will add this to the paper.
>
>  - Question: "How ...  predictive?"
>
>      - Answer: We believe that the gain is observable almost everywhere in practice, but the quality of the approximation fairly depends on how the assumptions on the paper are satisfied in practice. We will add more explanations about this limitation to the paper.

---

> > ### Comment · Reviewer_DGX1 · 2023-08-17
> >
> > Thank you for the thoughtful responses, I look forward to seeing the next draft!

---

### Official Review · Reviewer_doFr · 2023-07-06

**Soundness:** 3 good
**Presentation:** 3 good
**Contribution:** 3 good
**Rating:** 7
**Confidence:** 3

**Summary:**

The paper analyzes the generalization error of kernel ridge regression on manifolds with a kernel invariant to a group action. Compared with previous work, the results hold more generally for groups of positive dimension.


**Strengths:**

The paper tackles an important issue, is well-written, with well-chosen and appropriate examples, and appears technically sound. It also offers a differential geometric perspective on learning theoretic issues which might have its own interest.


**Weaknesses:**

Though already significant in its present form, the paper would gain in interest if it could provide more "agnostic" generalization error bounds in various ways, or at least comment on how (and if) this would be possible.

- Assumptions on f*: While interesting per se, the results are inapplicable to practical cases without knowledge of the target function f* (and the noise level sigma^2). In particular, Theorem 3.1 makes strong assumptions on the target function f* and holds only for the optimal value of the regularization parameter. It would be much stronger if it could be reformulated to hold more broadly for all regularization parameters with, e.g., the norm of \hat{f} instead of that of f*, and with f* only involved through the empirical risk (like more "traditional" error bounds based for instance on the Rademacher complexity).
In other words, if invariance to a (finite) group G intuitively implies that "each data point is worth |G| data points", does it also imply something like "the complexity of a model class taking into account the invariance is the standard complexity of the model class divided by (the square-root of) |G|" (where the complexity can refer for instance to the Rademacher complexity)?

- Assumptions on G: the results depend on knowledge of the group G. What if G is only approximately known? For instance, what if the kernel K is invariant to G' while f* is invariant to G with G' slightly different from G (or if G' contains only a subset of the transformations in G)?

- How much of the result relies on the uniform distribution of x on the manifold? Could you comment on what terms would be affected by a change of measure in Theorem 3.1 and if the gain as G grows remains of the same order in this case?


**Questions:**

No question

**Limitations:**

No limitation identified.

---

> ### Author Rebuttal · Authors · 2023-08-10
>
>
> We thank the respected reviewer for their helpful comments. Here we provide a response to the questions/weaknesses mentioned by the reviewer.
>
>
>
> - Question: "Assumptions on f*"
>
>     - Answer: We thank the reviewer for mentioning this important comment. Indeed, the KRR algorithm does not need any information about the function $f^\star$ rather than its Hilbert space norm $|| f^\star ||_{\mathcal{H}}$ (just to determine the regularization parameter). However, according to Equation (120) in the appendix, we only need an upper bound on the Hilbert space norm of the function $f^\star$ (which is a common assumption while studying kernels). The upper bound will let us compute the regularization parameter and run the algorithm without any further knowledge of the target function, and also Theorem 3.1 is still valid if one replaces the upper bound on the Hilbert space norm of the target function instead of its true value (same is true for the noise variance). We note that having an upper bound on the Hilbert space norm is essential to prove an upper bound on the population risk because the Hilbert space norm, intuitively, characterizes the "complexity" of the class of function to be learned. We thank the reviewer for mentioning this important aspect of the result and will add a detailed explanation in the next version of the paper.
>
>     Also, for scaling the complexity of the model class under invariances, we believe that the complexity will reduce as the reviewer correctly mentioned, but the exact scaling highly depends on the notion of the complexity used in the problem. We believe our dimension counting formula could be used as a  way to compactly quantify this complexity, and have potential applications to future works.
>
>
>
> - Question: "Assumptions on G"
>
>     - Answer: Since the results in the paper hold generally for arbitrary Lie groups, if one only knows the invariances according to a smaller group $G'$ then the upper bound is still valid and is reduced to the one with $G'$ replaced instead of $G$.  We will add explanations about this to the next version of the paper.
>
>
> - Question: "How much ..."
>
>     - Answer: Thanks for mentioning this comment. We cannot expect the gain to be true for "any" distribution. For example, think of a distribution supported on only a submanifold of the original manifold, then the problem is effectively learning on a smaller manifold and all the volume/dimensions must be corrected accordingly. However, if the sampling distribution has full support, with an upper bound or lower bound on its density (compared to the uniform distribution), then the same rate is achievable within constant factors. Since the main goal of this paper was to quantify the gain of "invariance" we just focused on the uniform sampling for simplicity while it's not restrictive. We will add these explanations to the paper in the next version. it's

---

### Official Review · Reviewer_xLSq · 2023-07-10

**Soundness:** 4 excellent
**Presentation:** 4 excellent
**Contribution:** 4 excellent
**Rating:** 8
**Confidence:** 3

**Summary:**

In this work the authors theoretically study how encoding invariances into models improves sample complexity. They approached the problem from differential geometric viewpoint, rather than common strategy of using invariant polynomials. Since the problem is algorithm and model dependent, the authors considered kernel-based algorithms, as neural networks in certain regimes behave like so. The obtained results generalize and greatly expand previous state-of-the-arts. Hence, the results provide a reduction in sample complexity that was not possible with previous assumptions. The paper also shows how these results transfer to popular invariant models, such as DeepSets, GNNs, PointNet, and SignNet.

**Strengths:**

- The paper discuss one of the major challenges in machine learning, i.e., sample complexity.
- This work provides theoretical results on how incorporating invariance in model can improve sample complexity. Their results are more general and can achieve better bound.
- Well written.

**Weaknesses:**

- None

**Questions:**

- None

**Limitations:**

- Theoretically intense, so for some readers, might be hard to follow.

---

> ### Author Rebuttal · Authors · 2023-08-10
>
>  We thank the respected reviewer for their positive review and appreciation of the theoretical results. For the next version, we will pass the paper and make it more readable to address the issue mentioned in the limitations section. Indeed, we are planning to add figures to the paper to make the problem and contributions more visible.

---

> > ### Comment · Reviewer_xLSq · 2023-08-18
> > **Response to rebuttal**
> >
> > I want to thank the authors for the rebuttal. I still think the paper addresses an important problem and the solution proposed is theoretically sound. I appreciate the authors intent to add figures in the main paper to help explaining the problem and contribution better. I will like to keep my rating as strong accept.

---

### Official Review · Reviewer_jgVU · 2023-07-30

**Soundness:** 4 excellent
**Presentation:** 3 good
**Contribution:** 4 excellent
**Rating:** 5
**Confidence:** 4

**Summary:**

This article investigates the sample complexity gained from encoding invariances into learning models. The article focuses on the study of kernel ridge regression on compact manifolds for functions that are invariants to a group action on the manifold. The main result of this article (Theorem 3.1) gives an upper bound of the excess population risk for functions living in the intersection of Sobolev spaces and the set of G-invariant square-integrable functions on the manifold. The analysis shows the importance of two terms: the volume of the quotient space, and the effective dimension of the quotient space. This result quantifies the exact sample complexity gain from invariances for kernel regression on compact manifolds for an arbitrary Lie group action. Moreover, the authors prove that the KRR estimator is minimax optimality. The proofs of the results make use of new results in the field of differential geometry.

**Strengths:**

- Although heavily technical, the article is well written and the proven results were extensively discussed.
- The newly proved results may be beneficial for the community of kernel methods on manifolds.

**Weaknesses:**

To be honest, I didn't have enough time to check the proofs thoroughly. Given that the contribution is purely theoretical, I believe it will be better suited for a math-oriented journal, where a rigorous scrutiny of the proofs can be ensured.


**Questions:**

-

**Limitations:**

-

---

> ### Author Rebuttal · Authors · 2023-08-10
>
> We thank the respected reviewer for their helpful comments. Here we provide a response to the questions/weaknesses mentioned by the reviewer.
>
>  - Question: "To be honest ... can be ensured."
>
>
>     - Answer: We appreciate the reviewer for mentioning this important comment.  While we cannot put the proofs into the paper (for page limit), we have provided a brief proof sketch in the main body of the paper (Section 3.1), as well as a more detailed proof sketch in Appendix A. We hope that the outline of the proof we provided is useful for the reader.
>
>
> Also, many theoretical papers appear in ML venues and we believe that theoretical papers have a place and are useful in the ML community (although this paper is applicable to kernel methods on manifolds, it is also applicable to kernel approximations of neural networks (e.g., in lazy regimes)). We hope that our theory can help better understand the effects of group invariances in the learning problem.

---

### Decision · Program_Chairs · 2023-09-21

**Decision:**

Accept (spotlight)

**Comment:**

The submission is about understanding the gain achievable when learning with invariances under the umbrella of reproducing kernel Hilbert spaces. More particularly, the authors focus on kernel ridge regression (2) on smooth connected compact Riemannian manifolds, where the target function is invariant w.r.t. compact Lie group actions. Using differential geometric tools, they establish minimax optimal (Theorem 3.3) and achievable (Theorem 3.1) excess risk bounds in this setting which show the gain terms of the reduction of the volume of the manifold and that of the effective dimension. The theoretical insights are followed by examples of learning scenarios with commonly-deployed invariances.

Kernel methods are among the most powerful tools of our times. The submission is clearly structured and well-written. The reviewers agreed on the novelty and importance of the theoretical results established in the manuscript. The paper is of definite interest to the NeurIPS community.